# PuzzlePlex: A Benchmark to Evaluate the Reasoning and Planning of Large Language Models on Puzzles

## Abstract

Large Language Models (LLMs) have demonstrated remarkable performance in various tasks, yet their comprehensive reasoning and planning capabilities in interactive environments remain underexplored. We introduce PuzzlePlex, a benchmark designed to evaluate reasoning and planning capabilities in a multi-turn competitive two-player environment. PuzzlePlex comprises 24 diverse puzzles, including deterministic and stochastic games, as well as single-player and competitive two-player scenarios. An important novelty of our benchmark is that it includes multi-step competitive two-player reasoning games. To succeed in such games, each LLM must maintain a history of its own moves and those of the opponent LLM, generating strategies that outperform the opponent to secure victory. We implement customized game-playing strategies (such as dynamic programming approaches) for comparison. Our findings indicate that the reasoning and planning abilities of current LLMs are currently poor in puzzle-solving contexts. GPT-4 outperforms other models, successfully competing against customized strategies (such as greedy approaches or dynamic programming) in 49% of cases. However, when faced with strict rule sets, it demonstrates diminished reasoning and planning capabilities. In addition to the 14 multi-turn competitive two-player puzzles, we report on single-player puzzles and incorporate multimodal challenges that integrate text and images, revealing that LLMs still significantly lag behind even simple heuristics in puzzles. A key feature of our benchmark is its ability to generate game instances with graduated levels of difficulty, allowing it to evolve as LLMs become more sophisticated. This adaptability ensures the continued relevance and utility of PuzzlePlex in assessing the progress of LLM capabilities in reasoning and planning within interactive environments.[1]

## 1 Introduction

Large language models (LLMs) have demonstrated performance comparable to humans in a range of tasks, from nuanced natural language understanding to complex math word problem solving (Team et al., 2023; Touvron et al., 2023). These capabilities highlight their potential not just as tools for automated responses but as shallow problem-solvers that can navigate and interpret extensive data sets with impressive accuracy, which is crucial for decision-making system.

This work uses puzzle solving as a means to enhance the evaluation of LLMs for deep problem-solving. By deep problem-solving, we mean a combination of logical and numerical reasoning, strategic planning, and adaptability. Puzzles possessing these characteristics are excellent candidates for assessing the capabilities and limitations of LLMs in scenarios that mimic complex real-world problem solving. While some prior research works have explored evaluating LLMs in puzzle-solving contexts (Noever & Burdick, 2021; Ding et al., 2023), they have focused on single-agent interactions with the environment (Stechly et al., 2024), where game information is typically conveyed solely through text descriptions of rules and states, and where each action leads to a predetermined, non-stochastic reward (Shridhar et al., 2020; Yao et al., 2022). PuzzlePlex, in contrast, focuses

---

[1]The code and data are available on `https://anonymous.4open.science/r/PuzzlePlex-224A/`.

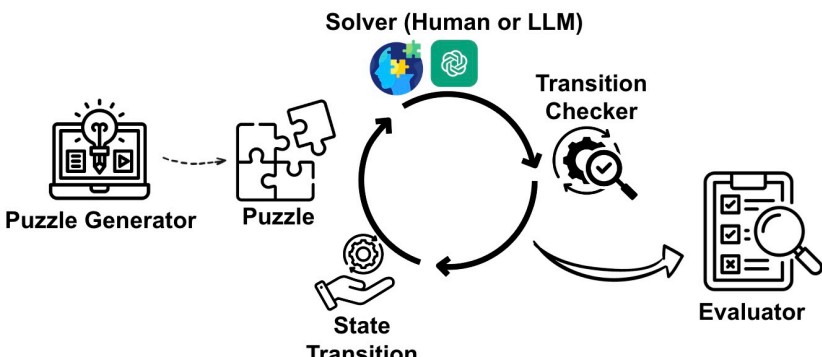

Figure 1: Overview of the developed pipeline framework. **Puzzle Generator** creates puzzle instances from templates based on the puzzle name, difficulty level, and selected competing models. The **Solver** then generates a response after receiving the puzzle instance. This response is passed to the **Transition Checker**, which verifies the legality of the operation output by the **Solver** and checks the game status. If the game ends, the **Evaluator** calculates and outputs the score. Otherwise, **State Transition** updates the state and passes the updated information back to the **Solver**.

more on reasoning and planning under uncertainty. Furthermore, PUZZLEPLEX is the first benchmark to evaluate the performance of LLMs in both multi-turn competitive two-player scenarios and multimodal settings of puzzles.

In addition, each puzzle in PUZZLEPLEX is available at multiple difficulty levels. We start with easy and intermediate levels. For single-player puzzles, difficulty is adjusted by varying the initialization size or slightly modifying the rules. For two-player puzzles, we provide baseline strategies for the LLM player to compete against, representing the two difficulty levels. The strategies we employ and the puzzle instance sizes are designed to stay within human cognitive limits.

The framework for our approach is presented in Figure 1and will be discussed in more detail in §3.1. Our contributions are as follows:

- We introduce PUZZLEPLEX, a benchmark dataset that includes 24 parametrizable mostly novel puzzles, including single-player, competitive two-player, deterministic, stochastic, text and text-image games. PUZZLEPLEX is the first benchmark to include multi-turn competitive two-player puzzles, thus requiring context-driven deep reasoning.
- Each puzzle in the dataset is accompanied by a generator that creates multiple instances at varying levels of difficulty. PUZZLEPLEX also provides a set of baseline strategies for solving each puzzle.
- Our framework enables both human and computational players to interact with the puzzles, capturing state transitions and recording output scores for comprehensive analysis.
- We conduct extensive experiments across a wide range of LLMs, providing a comprehensive, multi-dimensional assessment of their capabilities. The results reveal that current LLMs still face significant limitations in reasoning and planning within the context of game playing.

## 2 RELATED WORK

### 2.1 PUZZLES AND RELEVANT BENCHMARKS

Puzzles can be broadly categorized into rule-based and rule-less types. Rule-based puzzles, such as Sudoku (Noever & Burdick, 2021), Crosswords (Sadallah et al., 2024), and Chess (Feng et al., 2024), have well-defined victory conditions, permissible moves, and state transition rules. These games typically require strategic planning and logical reasoning. In contrast, rule-less puzzles like Riddles (Lin et al., 2021; Bisk et al., 2020; Zhang & Wan, 2022) lack predefined move sets or objectives. PUZZLEPLEX focuses exclusively on rule-based puzzles, allowing for an objective evaluation of LLMs capabilities in competitive scenarios. We prioritize puzzles that do not rely on world

Table 1: Comparison between PUZZLEPLEX with existing puzzle benchmarks. A single turn game is one in which the game ends after one move by one or more players. In the single player setting, an example is a pull of a slot machine. In the multiplayer competitive setting, an example is an instance of rock-paper-scissors. PUZZLEPLEX is one of the few benchmarks (along with SmartPlay (Wu et al., 2024)) that includes multi-turn games and the only one that includes multi-turn competitive two-player games. PUZZLEPLEX is also the only benchmark that allows text-image benchmarks.

| Benchmark | Game Scenario | | Reward Predictability | | # Multi-Turn | Data Type | | Varying Difficulty |
|---|---|---|---|---|---|---|---|---|
| | Single-player | Competitive Two-player | Deterministic | Stochastic | | Text | Text-Image | |
| PuzzleBench (Mittal et al., 2024) | ✓ | ✗ | ✓ | ✗ | ✗ | ✓ | ✗ | ✗ |
| LogicGame (Gui et al., 2024) | ✓ | ✗ | ✓ | ✗ | ✗ | ✓ | ✗ | ✓ |
| BoardgameQA (Kazemi et al., 2024) | ✓ | ✗ | ✓ | ✓ | ✗ | ✓ | ✗ | ✗ |
| P3 (Schuster et al., 2021) | ✓ | ✗ | ✓ | ✗ | ✗ | ✓ | ✗ | ✓ |
| PUZZLEQA (Zhao & Anderson, 2023) | ✓ | ✗ | ✓ | ✗ | ✗ | ✗ | ✗ | ✗ |
| SmartPlay (Wu et al., 2024) | ✓ | ✓ | ✓ | ✓ | ✓ | ✓ | ✗ | ✗ |
| **PUZZLEPLEX** (ours) | ✓ | ✓ | ✓ | ✓ | ✓ | ✓ | ✓ | ✓ |

knowledge, as those requiring extensive background information (Schuster et al., 2021; Lin et al., 2021; Todd et al., 2024) can be challenging for most humans. For instance, the puzzle *guess my city* (Abdulhai et al., 2023) can be challenging for individuals familiar with only a few major cities. However, due to their extensive training on global knowledge, state-of-the-art LLMs have already surpassed human performance in these knowledge-intensive tasks.

Table 1 presents a comparison between PUZZLEPLEX and other puzzle benchmarks. Based on different scenarios, puzzle benchmarks can be categorized into single-player puzzles (Mittal et al., 2024; Gui et al., 2024; Zhao & Anderson, 2023), which emphasize individual problem-solving skills and strategy development. Competitive two-player puzzle benchmarks involve competitive interactions between multiple agents, in a single-turn setting (Wu et al., 2024) using LLMs. However, in the domain of multi-turn competitive two-player games, only a limited number of benchmarks exist. AgentBench (Liu et al., 2023) includes one such game, but there is a notable lack of benchmarks that thoroughly address competitive two-player games. Furthermore, puzzles can be classified as either stochastic, which introduce elements of randomness, or deterministic, which depend solely on logical reasoning and player choices. Currently, there is a lack of benchmarks that encompass all these types, especially in competitive two-player scenarios under multi-turn settings. In addition, unlike existing benchmarks (Wu et al., 2024; Kazemi et al., 2024), PUZZLEPLEX also includes text-image puzzles, which challenge the abilities of LLMs in integrating information from different modalities through multi-turn interactions with the environment.

## 2.2 EVOLUTION OF PUZZLE SOLVING TECHNIQUES

A wide range of methods and strategies have been employed to solve rule-based puzzles. These include algorithmic techniques such as dynamic programming (Smith, 2007), alpha-beta pruning (Korf, 1990), and search algorithms (Lewis, 2007). In the domain of single-player games, neuro-symbolic approaches are popular (Ahmed et al., 2023; Murali et al., 2019) because single-player puzzles often require combinatorial search and can typically be reduced to Satisfiability (SAT) problems (Bright et al., 2020; Høfler, 2014). With advances in deep learning algorithms, reinforcement learning has become increasingly popular in solving puzzles (dos Santos et al., 2022; Huang et al., 2024). However, despite the enhanced computational power available today, the combinatorial explosion of many puzzles means that heuristic methods remain useful (Silver et al., 2016).

In the context of early LLMs, fine-tuning is often used to solve puzzles. For example, researchers have fine-tuned models such as GPT-2 (Radford et al., 2019) and FLAN-PaLM (Chung et al., 2024) to solve puzzles like Sudoku (Noever & Burdick, 2021) and BoardgameQA (Kazemi et al., 2024).

The advent of powerful LLMs (Achiam et al., 2023; Anthropic, 2024) has introduced a more flexible approach for solving puzzles through few-shot in-context learning. By translating puzzles into natural language descriptions and using language-based feedback in a multi-run setting, recent work has explored the capabilities of LLMs in puzzle solving. The Chain-of-Thought (CoT) approach (Wei et al., 2022) has proven superior to simple prompts in this context. Additionally, other prompting techniques have been successfully applied to puzzles. For example, Self-Refine (Madaan et al., 2024) is used for the Game of 24; Tree-of-Thought (Long, 2023) employs a tree structure to solve Sudoku; and Everything-of-Thought (Ding et al., 2023) utilizes graph topology to solve three de-

terministic puzzles. In our work, we employ prompting techniques similar to CoT to test whether current LLMs and MLLMs can effectively leverage their knowledge and understanding to reason comprehensively, plan, and make decisions when faced with complexity.

## 3 THE PUZZLEPLEX BENCHMARK

We first introduce the PUZZLEPLEX framework in which puzzle templates can be instantiated, moves recorded, state information shared, and states evaluated. We next describe the puzzles included in this benchmark, the implementation of baseline strategies, and the evaluation method.

### 3.1 PUZZLE GENERATION FRAMEWORK

PUZZLEPLEX has the following main components, as presented in Figure 1.

**Instance Generation**   For each puzzle $p$, we distinguish between a possibly parametrized puzzle template $template(p)$ (e.g., Sudoku on a $9 \times 9$ grid, template(Sudoku(9,9)), and an instance $instance(p)$ (e.g., a particular instance of Sudoku on a $9 \times 9$ grid, instance(Suduoku(9,9)). A generator function $G_p$ maps templates to instances. The generated instance is also the initial state $S_0$ of the game. That is, $instance(p) = S_0$. The generator for each puzzle will create instances using randomness, and it will adjust the difficulty level by varying the size of the puzzle.

**State Transition**   After receiving a move $M$ generated by a player (human or computer), the state transition module maps a state $S_n$ to a new state $S_{n+1}$ while incorporating feedback $F_n$. The feedback $F_n$ indicates the legality of the move, whether the game has terminated, and provides new position information. This process is represented as $M : S_n \rightarrow (S_{n+1}, F_n)$.

**Evaluation**   Once the puzzle-solving process terminates, an **evaluator** $E_p$ is applied to the sequence of states $S_0, S_1, \ldots, S_n$ to determine the raw score(s), represented as $rs_p = E_p(S_0, S_1, \ldots, S_n)$. The scale of the raw scores varies depending on the resolution type of each puzzle. To ensure comparability, we normalize these scores to obtain final scores ranging from 0 to 1 (§ 3.4).

To better keep track of state transitions and model reasoning steps, we implemented a Web UI called **Simulator** for visual observation. An example of this interface is shown in the § A.2.

### 3.2 PUZZLEPLEX BENCHMARK CONSTRUCTION

Aside from the four classical puzzles Sudoku, SudokuM (text-image version of Sudoku), N-Queens, and Takuzu, the puzzles in our study are derived from a column of Communications of the ACM [2]. While LLMs may have accessed the texts describing these puzzles, there are no strategies for them online, thus eliminating the possibility of data contamination. Additionally, we have simplified the rules for several puzzles to lower the barrier to entry, allowing most people to engage with them immediately after learning the rules and objectives.

Our 24 puzzles can be categorized into four types: **single-player deterministic**, **single-player stochastic**, **competitive two-player deterministic**, and **competitive two-player stochastic**. Text-based puzzles encompass all four types, while text-image puzzles are limited to single-player deterministic and competitive two-player deterministic variants. The distinction between deterministic and stochastic games lies in the predictability of operation rewards. In deterministic games, the outcome of a decision is fixed, regardless of how many times it is chosen. Conversely, stochastic games yield probabilistic outcomes, where repeated selection of the same operation in the same state may result in different outcomes. Detailed information about the puzzles is presented in §A.1, with individual puzzle descriptions provided in §A.3.

### 3.3 BASELINE STRATEGIES

We implemented baseline strategies for each puzzle, which can be categorized as follows:

---

[2]https://cacm.acm.org/section/opinion/

- Satisfiability Modulo Theories (SMT) Solver: This approach involves encoding puzzle constraints and rules as logical formulas, which are then solved using an SMT solver. SMT solvers determine whether a set of constraints is satisfiable and, if so, provide a solution that satisfies all constraints. In our customized strategies, we utilize the powerful Z3 Solver package (De Moura & Bjørner, 2008).
- Brute-force Algorithm: This method is employed when the problem size allows for an exhaustive search within our specified time constraints.
- Search Algorithms: We employ a variety of search techniques, including:
  - Uninformed search methods: Breadth-First Search (BFS) and Depth-First Search (DFS).
  - Probabilistic search: Monte Carlo Tree Search (MCTS).
- Dynamic Programming (DP): Dynamic programming is applied to puzzles that exhibit overlapping subproblems and optimal substructure.
- Greedy Algorithm: Greedy algorithms are employed in puzzles where locally optimal choices are expected to lead to globally optimal solutions or the search space is too large for other techniques, often reflecting strategies used in real-world scenarios.
- Other Methods: Additional algorithms, such as backtracking and simulated annealing algorithm, are incorporated.

**Single-player Games**    In this setting, we implemented one strategy for each puzzle, which remains consistent regardless of the difficulty level. This approach is justified because single-player games do not require competition with another player.

**Competitive Two-player Games**    We employ baseline strategies that vary based on the difficulty level, as they necessitate competition with another player. This variety of strategies allows for a more comprehensive evaluation of LLMs capabilities. At the easy level, baseline strategies for most puzzles employ legal random moves, selecting randomly from the space of legal moves. At the intermediate level, strategies become more sophisticated, occasionally identifying superior moves. These strategies may also consider moves that confer advantages over the opponent. The strategies utilized for each puzzle, along with examples of customized strategies, can be found in § B.2.

### 3.4 EVALUATION METRICS

Metrics for evaluating the performance of LLMs on puzzles can be characterized as either **binary** or **continuous**.

**Single-player Game**    In binary metric puzzles, players either succeed in achieving the desired outcome, resulting in a score of 1, or fail, yielding a score of 0. By contrast, continuous metric games involve the accumulation of points based on various factors such as number of moves, constraints, or objectives, leading to scores that may fall outside the [0, 1] interval. To ensure comparability across both types of games, we normalize raw scores onto a common scale of [0, 1]. This standardization process typically involves employing a baseline strategy with identical initialization parameters and utilizing its performance as a reference point. In cases where higher scores indicate better performance, if the score of the LLM model exceeds that of the baseline, it is assigned a score of 1; otherwise, the final score is determined by the ratio of the raw score of the LLM model to that of the baseline. This metric holds symmetrically in situations where lower scores signify superior performance.

**Competitive Two-player Games**    The metric for competitive two-player games is ternary because there are three final possible outcomes: win, lose, and tie. We assign scores to these outcomes: a score of 1 for a win, 0 for a loss or tie. However, in some such games, the order of play (being the first or second mover) can be advantageous. To account for this, we will run a game between two players A and B twice, once with player A as the first mover and once as the second mover, and then take the average of the scores obtained in both scenarios as the final score. This approach ensures that the final score is not biased by the order of play and provides a balanced assessment of the player's performance.

Table 2: Results for single-player scenario in text puzzles. FIR stands for **Failure Illegal Rate**, which represents the percentage of illegal moves made by a model that result in an immediate failure, even when a legal move is available.

| Model | Size | Deterministic Games | | | | Stochastic Games | | | | All Games | | | | Average | |
| | | Easy | | Inter. | | Easy | | Inter. | | Easy | | Inter. | | Score | FIR |
| | | Score | FIR | Score | FIR | Score | FIR | Score | FIR | Score | FIR | Score | FIR | | |
|---|---|---|---|---|---|---|---|---|---|---|---|---|---|---|---|
| Baseline | - | 1.00 | 0.00 | 1.00 | 0.00 | 1.00 | 0.00 | 1.00 | 0.00 | 1.00 | 0.00 | 1.00 | 0.00 | 1.00 | 0.00 |
| GPT-4o | - | **0.27** | **0.53** | **0.22** | 0.58 | 0.64 | **0.00** | 0.54 | **0.03** | **0.39** | 0.36 | 0.33 | 0.40 | **0.36** | **0.38** |
| GPT-3.5-turbo | - | 0.10 | 0.70 | 0.15 | 0.73 | 0.60 | 0.27 | 0.43 | 0.50 | 0.27 | 0.56 | 0.24 | 0.66 | 0.26 | 0.61 |
| Gemini-1.5-Pro | - | 0.22 | 0.67 | 0.20 | 0.67 | 0.62 | 0.13 | 0.50 | 0.30 | 0.35 | 0.49 | 0.30 | 0.54 | 0.33 | 0.52 |
| Gemini-1.5-Flash | - | 0.18 | 0.68 | 0.18 | 0.68 | 0.32 | 0.53 | 0.24 | 0.57 | 0.23 | 0.63 | 0.20 | 0.64 | 0.22 | 0.64 |
| Claude-3.5-Sonnet | - | 0.23 | 0.65 | 0.10 | 0.72 | 0.63 | 0.07 | 0.54 | 0.07 | 0.37 | 0.46 | 0.25 | 0.50 | 0.31 | 0.48 |
| Llama-3.1 | 405B | 0.15 | 0.72 | 0.13 | 0.75 | 0.39 | 0.37 | 0.45 | 0.33 | 0.23 | 0.60 | 0.24 | 0.61 | 0.24 | 0.61 |
| Llama-3.1 | 70B | 0.22 | 0.62 | 0.18 | **0.57** | 0.56 | 0.19 | 0.50 | 0.40 | 0.33 | 0.48 | 0.29 | 0.51 | 0.31 | 0.50 |
| Llama-3.1 | 8B | 0.03 | 0.82 | 0.02 | 0.83 | 0.62 | 0.17 | 0.58 | 0.28 | 0.23 | 0.60 | 0.21 | 0.65 | 0.22 | 0.63 |
| Mistral | 8×22B | 0.18 | 0.62 | 0.20 | 0.62 | 0.65 | 0.21 | 0.58 | 0.36 | 0.34 | 0.48 | **0.33** | 0.53 | 0.34 | 0.51 |
| Mistral | 8×7B | 0.15 | 0.65 | 0.17 | 0.65 | 0.57 | 0.35 | 0.54 | 0.42 | 0.29 | 0.55 | 0.29 | 0.57 | 0.29 | 0.56 |
| Qwen2 | 72B | 0.18 | 0.70 | 0.17 | 0.73 | **0.66** | 0.15 | **0.62** | 0.22 | 0.34 | 0.52 | 0.32 | 0.56 | 0.33 | 0.54 |

Table 3: Results of win-fractions comparing each model to the baseline strategy on competitive two-player multi-turn deterministic and stochastic games at two different difficulty levels. GPT-4o did best on deterministic games and on intermediate stochastic games, while Claude 3.5-Sonnet did very well on easy Stochastic Games.

| Model | Size | Deterministic Games | | | | Stochastic Games | | | | All Games | | | | Average | |
| | | Easy | | Inter. | | Easy | | Inter. | | Easy | | Inter. | | Score | FIR |
| | | Score | FIR | Score | FIR | Score | FIR | Score | FIR | Score | FIR | Score | FIR | | |
|---|---|---|---|---|---|---|---|---|---|---|---|---|---|---|---|
| GPT-4o | - | **0.56** | **0.21** | **0.43** | 0.22 | 0.45 | **0.00** | 0.45 | **0.00** | **0.54** | 0.18 | **0.43** | 0.18 | **0.49** | **0.18** |
| GPT-3.5-turbo | - | 0.32 | 0.42 | 0.26 | 0.41 | 0.45 | **0.00** | 0.35 | 0.05 | 0.34 | 0.35 | 0.28 | 0.35 | 0.31 | 0.35 |
| Gemini-1.5-Pro | - | 0.37 | 0.29 | 0.37 | 0.28 | 0.60 | **0.00** | 0.40 | 0.05 | 0.41 | 0.24 | 0.38 | 0.24 | 0.40 | 0.24 |
| Gemini-1.5-Flash | - | 0.32 | 0.28 | 0.36 | 0.28 | 0.25 | 0.05 | 0.45 | **0.00** | 0.31 | 0.24 | 0.38 | 0.23 | 0.35 | 0.24 |
| Claude-3.5-Sonnet | - | 0.43 | 0.26 | 0.37 | 0.24 | **0.70** | **0.00** | 0.30 | **0.00** | 0.48 | 0.22 | 0.36 | 0.20 | 0.42 | 0.21 |
| Llama-3.1 | 405B | 0.39 | 0.32 | 0.36 | 0.26 | 0.40 | 0.15 | 0.30 | 0.25 | 0.39 | 0.29 | 0.35 | 0.26 | 0.37 | 0.28 |
| Llama-3.1 | 70B | 0.40 | 0.25 | 0.40 | 0.26 | 0.51 | 0.01 | 0.27 | **0.00** | 0.42 | 0.21 | 0.38 | 0.22 | 0.40 | 0.22 |
| Llama-3.1 | 8B | 0.27 | 0.56 | 0.20 | 0.56 | 0.38 | 0.40 | 0.15 | 0.33 | 0.29 | 0.53 | 0.19 | 0.52 | 0.24 | 0.53 |
| Mistral | 8×22B | 0.33 | 0.41 | 0.34 | 0.28 | 0.40 | 0.15 | 0.17 | 0.17 | 0.34 | 0.37 | 0.31 | 0.26 | 0.33 | 0.31 |
| Mistral | 8×7B | 0.24 | 0.39 | 0.28 | 0.38 | 0.36 | 0.24 | 0.16 | 0.28 | 0.26 | 0.37 | 0.26 | 0.36 | 0.26 | 0.32 |
| Qwen2 | 72B | 0.41 | 0.29 | 0.41 | 0.23 | 0.52 | 0.02 | 0.30 | 0.03 | 0.43 | 0.24 | 0.39 | 0.20 | 0.41 | 0.22 |

Table 4: Results for the setting where the LLMs are provided with the legal moves show that GPT-4 demonstrates the best reasoning ability when a list of legal moves is included in the prompt. Surprisingly, some LLMs choose illegal moves even when legal ones are given.

| Model | Size | Sudoku | | SudoKill | |
| | | Score | FIR | Score | FIR |
|---|---|---|---|---|---|
| GPT-4o | - | **0.50** | **0.50** | **0.70** | **0.30** |
| GPT-3.5-Turbo | - | 0.10 | 0.90 | 0.00 | 1.00 |
| Gemini-1.5-Pro | - | 0.00 | 1.00 | 0.60 | 0.40 |
| Gemini-1.5-Flash | - | 0.30 | 0.70 | 0.60 | 0.40 |
| Claude-3.5-Sonnet | - | 0.30 | 0.70 | 0.40 | 0.60 |
| Llama-3.1 | 405B | 0.10 | 0.90 | **0.70** | **0.30** |
| Llama-3.1 | 70B | 0.10 | 0.90 | **0.70** | **0.30** |
| Llama-3.1 | 8B | 0.20 | 0.80 | 0.40 | 0.60 |
| Mistral | 8×22B | 0.20 | 0.80 | 0.40 | 0.60 |
| Mistral | 8×7B | 0.30 | 0.70 | 0.00 | 1.00 |
| Qwen2 | 72B | 0.40 | 0.60 | 0.50 | 0.50 |

**Strength** To better compare models in both single-player and competitive two-player games, we borrow the concept from the Bradley-Terry model (Hunter, 2004) and use a notion of **strength** to unify scores across different types of games. Because the Bradley-Terry model does not account for ties, we adopt the Davidson (Davidson, 1970) variant of the model. For single-player games, where direct pairwise comparisons are not naturally available, we construct pairs by comparing the scores of every two different models. The results are illustrated in the §C.1.

| Model | Win Rate |
|---|---|
| Claude-3.5-sonnet | 0.65 |
| Qwen2-72b | 0.61 |
| GPT-4o | 0.59 |
| Llama-3.1-70b | 0.59 |
| Llama-3.1-405b | 0.51 |
| Mistral-8x22b | 0.47 |
| Gemini-1.5-flash | 0.47 |
| Gemini-1.5-pro | 0.45 |
| GPT-3.5-turbo | 0.39 |
| Mistral-8x7b | 0.37 |
| Llama-3.1-8b | 0.35 |

Figure 2: The figure on the left comparing the results of competitive two-player multi-turn deterministic and stochastic games between pairs of models. Location $(i, j)$ shows the win percentage of model $i$ when playing against model $j$. The table on the right demonstrates the **Win Rate** of models sorted in descending order.

Table 5: Results of GPT-4 and Qwen2-72B using Tree-of-Thought (ToT) and 1-shot prompting across 4 puzzles when compared with CoT. For single-player games, we report scores at two difficulty levels. For competitive two-player games, we report performance when competing against both custom-designed methods at two difficulty levels and against other LLMs. For example, GPT-4o when using ToT wins 69% of the time against CoT in easy versions of Exclusivity Probes.

| Method | Sudoku | | SudoKill | | | Exclusivity Probes | | Larger Target | | |
|---|---|---|---|---|---|---|---|---|---|---|
| | Easy | Inter. | vs. Custom(E) | vs. Custom(I) | vs. LLMs | Easy | Inter. | vs. Custom(E) | vs. Custom(I) | vs. LLMs |
| GPT-4o | 0.10 | 0.00 | 0.00 | 0.00 | 0.66 | 0.12 | 0.03 | 0.70 | 0.50 | 0.61 |
| _w_. ToT | 0.40 (+0.30) | 0.00 | 0.10 (+0.10) | 0.00 | 0.64 (-0.02) | 0.69 (+0.57) | 0.11 (+0.08) | 0.60 (-0.10) | 0.40 (-0.10) | 0.64 (+0.03) |
| _w_. 1-shot | 0.20 (+0.10) | 0.00 | 0.00 | 0.00 | 0.67 (+0.01) | 0.20 (+0.08) | 0.05 (+0.02) | 0.50 (-0.20) | 0.50 | 0.63 (+0.02) |
| Qwen2-72B | 0.00 | 0.00 | 0.00 | 0.00 | 0.65 | 0.08 | 0.03 | 0.50 | 0.33 | 0.50 |
| _w_. ToT | 0.00 | 0.00 | 0.00 | 0.00 | 0.73 (+0.08) | 0.33 (+0.25) | 0.09 (+0.06) | 0.53 (+0.03) | 0.41 (+0.08) | 0.56 (+0.06) |
| _w_. 1-shot | 0.00 | 0.00 | 0.00 | 0.00 | 0.70 (+0.05) | 0.12 (+0.04) | 0.04 (+0.01) | 0.55 (+0.05) | 0.38 (+0.05) | 0.54 (+0.04) |

# 4 EXPERIMENTS

## 4.1 MODELS UNDER COMPARISON

The LLMs we evaluate include GPT-4o* (Achiam et al., 2023), GPT-3.5-turbo* (OpenAI, 2022), Gemini 1.5 Pro* (Reid et al., 2024), Gemini 1.5 Flash* (Reid et al., 2024), Claude 3.5 Sonnet* (Anthropic, 2024), Llama 3.1 (405B, 70B, 8B) (Meta, 2024a), Mistral (8x7B (Jiang et al., 2024), 8x22B (Mistral, 2024a)), and Qwen 2 (72B) (qwe, 2024). For text-image version puzzles, we evaluate on models includes GPT-4o*, Gemini 1.5 Pro*, Gemini 1.5 Flash*, Claude 3.5 Sonnet*, Pixtral (Mistral, 2024b) and Llama 3.2 Vision (11B, 90B) (Meta, 2024b). [3] We employ the chat or instruct versions of each model, as solving most puzzles requires multiple interaction rounds.

---

[3]Models with asterisks (*) superscripted are proprietary models.

## 4.2 EXPERIMENTAL SETUP

For single-player deterministic games, both the baseline strategies and the LLMs are tested on 10 instances, with random seeds set from 1 to 10 for reproducibility. For single-player stochastic games, we run 100 instances for custom methods and open-source models, except for Llama 3.1 405B[4].

For two-player games, the experiments are divided into baseline vs. LLMs and LLMs vs. LLMs. In the baseline vs. LLMs part, each strategy is tested on 5 instances with random seeds from 1 to 5, repeated twice to alternate the first player, and evaluated on two difficulty levels. For stochastic games involving custom vs. LLM matchups (except those involving Llama 3.1 405B), we increase the number of random seeds to 50 to ensure statistical significance. In the LLMs vs. LLMs part, 5 instances are tested for deterministic games, and 50 instances are used for stochastic games. In this setting, we do not vary difficulty levels.

To mitigate the risk of exceeding the contextual length, given the likelihood of multiple turns in our games, our evaluation primarily adopts a zero-shot CoT approach. The statistics of LLMs outputs for each puzzle is illustrated at §C.3.

## 4.3 MAIN RESULTS

For the single-player scenario text puzzles, we can see the results in Table 2. In the multi-turn competitive two-player scenario text puzzles, Table 3 shows the results of comparing an LLM against our customized method at two difficulty levels. Additionally, Figure 2 illustrates the results of LLM versus LLM competition. For text-image puzzles, Table 14 shows the result of single-player scenario, and Table 15 illustrates the result of competitive two-player scenario. Figure 29 illustrates the results of multi-turn competitive two-player scenario. Result statistics is provided at §C.2.

**Single-player Text Puzzles**    Table 2 shows that GPT-4o outperforms other models, yet all models significantly lag behind human performance, particularly in deterministic games. This disparity is largely attributable to the nature of deterministic games, which typically have less freedom and more stringent rules. Violating these rules often results in immediate game loss, as evidenced by the FIR exceeding 50% in deterministic games. This high FIR suggests that more than half of the trials failed due to illegal moves. Such results demonstrate that current LLMs, including state-of-the-art ones, still struggle with puzzle comprehension and identifying legal moves, indicating limited reasoning capabilities in game contexts. In contrast, stochastic games, with their less rigid rules, exhibit a considerably lower FIR than deterministic games.

**Competitive Two-player Text Puzzles**    Table 3 demonstrates that GPT-4o exhibits superior performance, with a win rate exceeding 50% in easy-level deterministic games, while approaching 50% in the other three settings. This can be partially attributed to the higher proportion of board games in PUZZLEPLEX single-player scenarios, which typically have larger state spaces and a smaller fraction of legal states. Correspondingly, the FIR of LLMs in deterministic games significantly surpasses that in stochastic games. In stochastic games, most LLMs exhibit an FIR close to zero, indicating their ability to comprehend game descriptions and adhere to system-requested output formats. However, their reasoning and planning capabilities for generating optimal or even legal moves remain a challenge. In LLM vs. LLM settings, Claude-3.5-Sonnet demonstrates the best performance, with open-source models showing comparable results to proprietary ones.

**Text-image Puzzles**    Current LLMs perform poorly on image data. One reason is that multimodal LLMs are typically trained on text-image pairs, without fine-tuning or training for multi-turn interactions. In these puzzles, the process involves multiple turns, each combining image and text, with images representing the game state. In *SudokuM* and *SudoKillM*, no model successfully completes or wins any instance, primarily due to strict rules, which is the same as text puzzles. *SuperplyM* has looser rules, allowing illegal moves without immediate game loss, alternating between players until the number of illegal moves exceeds a threshold or a player achieves the winning goal. The main challenge in this game is information extraction from images; LLMs struggle to accurately extract matrix information, hindering numerical reasoning.

---

[4]Due to budget constraints and computational limitations, we maintain the instances of 5 in deterministic games and the instances of 10 in stochastic games for Llama 3.1 405B and all proprietary models.

### 4.4 ANALYSIS OF PROMPTING STRATEGIES

Previous research demonstrates that few-shot learning (Min et al., 2022) and advanced prompting techniques, such as Tree-of-Thought (ToT) (Yao et al., 2024), can improve performance. Therefore, we applied these two prompting strategies to four puzzles, one from each category: *Sudoku* (single-player deterministic), *SudoKill* (competitive two-player deterministic), *Exclusivity Probes* (single-player stochastic), and *Larger Target* (competitive two-player stochastic). For few-shot learning, we opted for one-shot prompting due to the extensive length of gameplay, as evidenced by the average token count per game shown in the §C.3. For ToT implementation, we employed a sample strategy to generate five candidates for each thought step and utilized a voting strategy to evaluate states. We evaluated the performance using two representative models: GPT-4o and Qwen2-72B. The results are presented in Table 5. The data shows that both ToT and 1-shot prompting improve performance in most cases, with ToT yielding greater improvements than 1-shot prompting, and GPT-4o getting larger improvements than Qwen2-72B. However, these advanced prompting strategies are most effective when the state space is small. As the state space grows large, their impact becomes minimal. This suggests that while advanced prompting techniques enhance reasoning and planning abilities in small-scale games, their benefits do not scale effectively to larger, more complex scenarios.[5]

### 4.5 ANALYSIS OF REASONING AND PLANNING ABILITIES

In Section 4.3, we observed LLMs' limited reasoning abilities in identifying legal moves within state spaces. To explain their limitations, we investigate LLMs' reasoning and planning capabilities when provided with a list of candidate legal moves. We aim to determine if LLMs can effectively reason about these moves and select optimal strategies to increase their chances of winning against opponents.

For this analysis, we focus on two games with strict rules: *Sudoku* and *SudoKill*. In *Sudoku*, we utilize an easy-level $4 \times 4$ grid. For *SudoKill*, we limit the maximum length of the legal move list to 100 and pit the LLM against a baseline strategy that randomly selects moves from the provided list. By supplying legal moves, we enable LLMs to concentrate on planning without the burden of move identification.

Table 4 shows that GPT-4o performs best, although all models exhibit limited reasoning and planning abilities even when provided with legal moves. In Sudoku, most failures stem from inadequate planning; LLMs tend to randomly select legal moves from the provided list rather than employing foresight. For GPT-3.5-turbo and Mistral ($8 \times 7B$), in *SudoKill*, they lose all games due to their inability to follow instructions. Despite being provided with a list of legal moves, they often generate moves not in the list, resulting in game losses.

Therefore, the reasoning and planning abilities of current LLMs on puzzles remain limited. This limitation is evident from the high Failure Illegal Rate (FIR) in §4.3 and the low percentage of legal plays on puzzles requiring a through understanding of rules, as supported by the statistics in §C.3. These challenges arise because even small mistakes can lead to significant consequences, akin to a "butterfly effect," as discussed in §A.1. Moreover, LLMs struggle to generate effective plans even when provided with candidate moves.

### 4.6 ERROR ANALYSIS

To assess the capabilities and limitations of current LLMs on PUZZLEPLEX, we conducted an extensive error analysis, based on samples of 100 runs for each text games and 50 for each text-image games. Our analysis revealed four common error types (the definition of each type is demonstrated at Table 16):

**Reasoning and Planning Errors (63%)** These are the most common errors, as previously discussed. They occur when models fail to identify legal moves within the state space or lack effective planning to select advantageous moves. Such errors are more prevalent in games with more constrained rules. An example of this error shows at Figure 30. In addition, we observe instances of

---

[5]In the game of *Larger Target*, GPT-4o's performance compared to customized methods decreases as this is a stochastic game. Due to budget constraints, we were only able to run each setting 10 times, which limits the statistical significance.

faulty reasoning even when LLMs successfully solve puzzles or win against other LLMs. While these models may produce solutions that adhere to the required format of the game, their reasoning steps often contain errors. Among the samples we analyzed, we found that in 76% of cases where LLMs successfully solved puzzles or won against other LLMs, their success was not due to genuine reasoning. Instead, it was often the result of random moves (in games with loose rules) or the opponent making a critical mistake, such as a losing move, as illustrated in Figure 30. We also identified similar patterns across most LLMs in board games. For example, in games like *Sudoku*, *SudoKill*, and *Takuzu*, most LLMs (except for Claude-3.5-Sonnet) adopt a rigid approach. They typically fill cells sequentially, starting from the first empty cell in the top-left corner and progressing row by row. This approach disregards opportunities for moves that could lead to better outcomes. For instance, in a game of *Sudoku*, there might be a row (not the first row) that is almost complete, missing only one cell. Reasoning through this would provide an immediate solution for that row, but only Claude-3.5-Sonnet seems capable of prioritizing such an optimal strategy. An example is illustrated in Figure 32.

**Comprehension Errors (12%)** Because most games in PUZZLEPLEX are novel, LLMs lack prior exposure to their specific corpus or strategies. This makes them ideal for testing whether LLMs truly understand the rules. The outputs of several puzzles reveal that LLMs still face challenges in language comprehension. Although proprietary models generally perform better, we observe that, in some cases, open-sourced models demonstrate a better understanding of the rules compared to proprietary models. For example, in the games of *Max Target* and *Larger Target*, only the Llama-3.1-405B model recognizes that the provided bags are randomized, enabling them to exploit the game mechanics. An example illustrating this error is shown in Figure 33.

**Memorization Errors (11%)** These errors frequently occur in multi-turn scenarios, particularly in games with less rigid rules. After several steps, LLMs may lose track of previously visited states, leading to repetitive actions. For example, in *Exclusivity Probes*, all models repeatedly revisit positions they have already explored. This results in significantly more probes being required to find all the particles. As shown in §C.3, the average number of turns in this game is 23.10.

**Perception Errors (7%)** These errors indicate that models perceive the wrong state, which is a common issue in text-image puzzles. They typically occur when multimodal LLMs fail to accurately extract the game state from images, resulting in incorrect reasoning based on misinterpreted information.

**Other Errors (7%)** Other errors mainly involve failing to follow instructions, even when the LLMs make legal moves. For example, in the game of *Superply*, which combines numerical and spatial reasoning, models often focus solely on numerical reasoning. They identify a position that satisfies the hint but overlook that the position is already filled. Furthermore, even after receiving feedback, the models fail to recognize this mistake, leading to repeated errors, which is demonstrated at Figure 35.

## 5 CONCLUSION

PUZZLEPLEX is a benchmark focused on reasoning in many different puzzle and game settings: single-player/multi-player competitive; single-turn/multi-turn; deterministic/stochastic; and text-only/text-image. As far as we know, no other benchmark on LLMs includes either multi-player competitive or text-image puzzles. Multi-turn competitive games require the ability to evaluate a continually updating state. Including images enables LLMs to find visual patterns that may be obscure in text (e.g., in Sudoku).

A second major feature of PUZZLEPLEX is the provision of classic game-playing and puzzle-solving baseline techniques against which to compare LLMs over time.

The final major feature of PUZZLEPLEX is the ability to generate instances at graduated levels of difficulty, thus enabling the research community to conduct contests of increasing difficulty over time as LLMs improve.

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

## CONTENTS

# A PUZZLEPLEX

## A.1 DATASET OVERVIEW

Table 6: Overview of Puzzle Games and Their Basic Strategies. The column **butterfly-effect** describes whether the puzzle's rules are strict, requiring the model to have a clear understanding of the rules and the ability to identify illegal moves, as such a small move can lead to significant consequence. For example, in the game *SudoKill*, if a player fills a value in an illegal cell, the game immediately terminates, and the player loses. In contrast, in the game *Superply*, selecting the wrong cell simply turns the play over to the opponent without severe consequences.

| Name | Scenario | Reward | Data | Main Reasoning | Butterfly-effect |
|---|---|---|---|---|---|
| Sudoku (Wikipedia, 2024a) | Single-player | Deterministic | Text | Logical | ✓ |
| SudoKill | Competitive Two-player | Deterministic | Text | Logical | ✓ |
| Tidy Tower (Shasha, 2023) | Single-player | Deterministic | Text | Spatial | ✓ |
| Card Nim (Shasha, 2022a) | Competitive Two-player | Deterministic | Text | Numerical, Logical | ✓ |
| Expanding Nim | Competitive Two-player | Deterministic | Text | Numerical, Logical | ✗ |
| Share Card Nim | Competitive Two-player | Deterministic | Text | Numerical, Logical | ✗ |
| Optimal Touring | Single-player | Deterministic | Text | Numerical, Spatial | ✗ |
| Count Maximal Cocktails (Shasha, 2022c) | Single-player | Deterministic | Text | Logical | ✗ |
| Max Maximal Cocktails | Competitive Two-player | Deterministic | Text | Logical | ✓ |
| Exclusivity Particles (Shasha, 2022b) | Competitive Two-player | Deterministic | Text | Numerical, Spatial | ✓ |
| Exclusivity Probes | Single-player | Stochastic | Text | Numerical, Spatial | ✗ |
| Ruby Risks (Shasha, 2017) | Single-player | Stochastic | Text | Numerical, Logical | ✗ |
| Beat Or Bomb Det. | Competitive Two-player | Deterministic | Text | Logical, Numerical | ✗ |
| Beat Or Bomb Sto. | Competitive Two-player | Stochastic | Text | Logical, Numerical | ✗ |
| Max Target | Single-player | Stochastic | Text | Logical, Numerical | ✗ |
| Larger Target | Competitive Two-player | Stochastic | Text | Logical, Numerical | ✗ |
| Takuzu (Wikipedia, 2024b) | Single-player | Deterministic | Text | Logical | ✓ |
| KQueens | Single-player | Deterministic | Text | Logical | ✓ |
| Bid That | Competitive Two-player | Deterministic | Text | Logical, Numerical | ✗ |
| Bit That Vickrey | Competitive Two-player | Deterministic | Text | Logical, Numerical | ✗ |
| Superply | Competitive Two-player | Deterministic | Text | Numerical, Spatial | ✗ |
| Sudoku M. | Single-player | Deterministic | Text-Image | Visual, Logical | ✓ |
| SudoKill M. | Competitive Two-player | Deterministic | Text-Image | Visual, Logical | ✓ |
| Superply M. | Competitive Two-player | Deterministic | Text-Image | Visual, Numerical | ✗ |

## A.2 EXAMPLE OF SIMULATOR

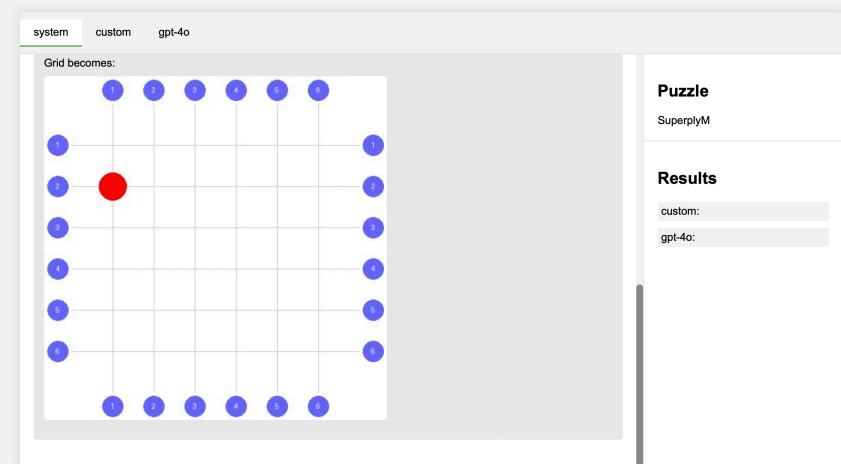

Figure 3: Overview of **Simulator**.

## A.3 Breakdown Description of Puzzles

---

**Sudoku**

You are given a grid of size `grid_size` × `grid_size`. The goal is to fill the grid with numbers such that each row, each column, and each of the subgrids (box) contains all of the numbers from 1 to `grid_size` without repetition. The grid has some cells filled with numbers already. You need to fill the empty cells, which are represented by 0. The input is a 2D list of integers representing the grid, with a 0-based index. At each time, you should only fill one empty cell.

For example, if the grid is

```
[[0, 3, 1, 2],
[1, 0, 4, 3],
[2, 1, 0, 4],
[3, 4, 2, 0]]
```
when you fill the cell (0, 0) with value 4, the grid becomes

```
[[4, 3, 1, 2],
[1, 0, 4, 3],
[2, 1, 0, 4],
[3, 4, 2, 0]]
```
Now please solve this grid: `sudoku_instance`.

**Raw Score:**
Success: 1, Failure: 0

**Type:**
Single-player - Deterministic

---

Figure 4: Description of Sudoku.

## SudoKill

The game is a 2-player twist on the classic Sudoku game. As in a traditional Sudoku game, you are given a 9x9 grid. The goal is to fill the grid with numbers so that each row, column, and 3x3 subgrid contains all numbers from 1 to 9 without repetition.

In Sudokill, the additional rule for this two-player game is: Players alternate placing numbers on the board. The first player can place a number in any unoccupied space. After that, each player must place their number in an unoccupied space in either the same row or column as the last move. If there are no such available spaces, the player can place a number anywhere on the board. The first player to make a move that violates the rules loses.

The grid has some cells pre-filled with numbers. Unoccupied cells are represented by 0. The input is a 2D list of integers representing the grid, using 0-based indexing. At each turn, you should fill only one empty cell.

For example, if the current grid is

```
[[6, 8, 4, 5, 1, 3, 2, 7, 9],
[5, 9, 7, 6, 2, 0, 1, 8, 0],
[2, 3, 1, 4, 8, 7, 6, 5, 0],
[9, 1, 2, 7, 6, 4, 8, 0, 3],
[4, 6, 8, 3, 0, 1, 7, 2, 5],
[7, 5, 3, 2, 9, 8, 4, 1, 6],
[8, 4, 5, 1, 3, 2, 9, 6, 7],
[1, 0, 6, 9, 0, 5, 0, 3, 8],
[3, 2, 0, 0, 7, 0, 5, 4, 0]]
```

and now is your turn and the previous move by the opponent is to fill the cell at (0, 8) with the value 9. So now the cells you can place a number are [(1,8), (2,8), (8,8)] because you can only place a number in the same row or column as the last move.

For example, if the current grid is

```
[[6, 8, 4, 5, 1, 3, 2, 7, 9],
[5, 9, 7, 6, 2, 0, 1, 8, 0],
[2, 3, 1, 4, 8, 7, 6, 5, 0],
[9, 1, 2, 7, 6, 4, 8, 0, 3],
[4, 6, 8, 3, 0, 1, 7, 2, 5],
[7, 5, 3, 2, 9, 8, 4, 1, 6],
[8, 4, 5, 1, 3, 2, 9, 6, 7],
[1, 0, 6, 9, 0, 5, 0, 3, 8],
[3, 2, 0, 0, 7, 0, 5, 4, 1]]
```

and now is your turn and the previous move by the opponent is to fill the cell at (0, 8) with the value 9. Now you can fill the cell (1, 8) with the value 4 to win this game because after you fill the cell (1, 8) with the value 4, the opponent can only fill the cell (2, 8) and (1, 5), but no matter which value the opponent fills in these two cells will violate the rules.

The initial grid is `sudokill_instance`.

**Raw Score:**
Win: 1, Lose: 0

**Type:**
Competitive Two-player - Deterministic

Figure 5: Description of SudoKill.

**TidyTower**

Your task is to solve a puzzle named 'Tidy Tower'. You are given a tower consisting of cubes, each of which has one of 4 colors. The goal is to align all cubes so that each color is the same vertically. A tower with such an alignment is called tidy.
Two kinds of operations are allowed:
1. Rotate a cube: Rotate a single cube, and all cubes above it rotate as well.
2. Rotate with holding: Rotate a cube and hold a cube above it, preventing it and the cubes above it from rotating.
The cube colors are represented by the letters R, Y, B, and G, corresponding to red, yellow, blue, and green respectively. The forward-facing side of each cube is indicated by the first letter in the sequence. The color sequence is in clockwise order.
Holding '0' means holding the cube above it, '1' means not holding the cube above it.

Here is an example:
Question 1: In this question, I indicate the forward-facing side with R, G, B, Y representing red, green, blue, and yellow respectively where the leftmost cube corresponds to the bottom cube (position 0): RGBYRGBYBGBGBG. Can you make this tower tidy in eight moves or less?
Solution for eight moves:
RGBYRGBYBGBGBG → (rotate by one position at position 1 and hold at position 2) RRGBYRGB-GRGRGR
→ (rotate by one position at position 2 and hold at position 3) RRRBYRGBGRGRGR
→ (rotate by two positions at position 3 and hold at position 4) RRRRYRGBGRGRGR
→ (rotate by one at position 4 and hold at position 5)RRRRRRGBGRGRGR
→ (rotate by one at position 6 and hold at position 9) RRRRRRGRRGRGR
→ (rotate by one at position 7 and hold at position 8)RRRRRRRRRGRGR
→ (rotate by one at position 10 and hold at position 11) RRRRRRRRRRRGR
→ (rotate by one at position 12 and hold at position 13) RRRRRRRRRRRRRR
Now please solve these cubes: `TidyTower_instance`.

**Raw Score:**
Success: 1, Failure: 0

**Type:**
Single-player - Deterministic

Figure 6: Description of TidyTower.

**CardNim**

In a game called Card Nim, each player has a collection of cards, each with a number on it. In each turn a player reveals a card and removes a number of stones equal to the number on the card. To win on a move, a player must play a card whose number is equal to the number of stones remaining.
And during the game, you can only play a card with a number that is less than or equal to the number of stones remaining.
For example, suppose there are five stones left and each of the two players you and your opponent has three cards with 1, 2,and 3, respectively.
You goes first. Who wins? Solution: Your opponent wins.
If you removes 2 or 3, then opponent can win immediately with 3 or 2 respectively.
So, you removes 1.Now your opponent removes 3, leaving 1. Now you has only cards with numbers greater than 1 so you lose.
Now please play on: `CardNim_instance`.

**Raw Score:**
Win: 1, Lose: 0

**Type:**
Competitive Two-player - Deterministic

Figure 7: Description of Card Nim.

1080
1081
1082
1083
1084
1085
1086
1087
1088
1089
1090
1091
1092
1093
1094
1095
1096
1097
1098
1099
1100
1101
1102
1103
1104
1105
1106
1107
1108
1109
1110
1111
1112
1113
1114
1115
1116
1117
1118
1119
1120
1121
1122
1123
1124
1125
1126
1127
1128
1129
1130
1131
1132
1133

### ExpandingNim

You need to play a game named "Expanding Nim" against your opponents.
And the rule is:
Initial number of stones: The game starts with a pile of stones, the number is determined by the game organizer.
Player Action: The first player may remove 1 to 3 stones.
The maximum number of removals for a subsequent player is determined by the number of removals for the previous player, called currentmax. Initially currentmax is 0.
If the reset option was used in the previous turn, the current player can remove 1 to 3 stones, otherwise the current player can remove up to (currentmax + 1) stone, and no more than 3 stones. Reset Options: Each player can use up to one reset option in the game. After using the reset option, players on the next turn can only remove up to 3 stones.
Game goal: The team that removes the last stone wins.
The reset option has a limited number of uses (maximum four per player), so it needs to be used at critical moments to gain an advantage.

Examples of game rules
Assume you start with 8 stones:
The first player can remove 1, 2 or 3 stones.
Suppose the first player removes 1 stone and now there are 7 stones remaining and currentmax is updated to 1.
The second player can remove up to 2 stones (currentmax + 1), assuming 2 stones are removed and now there are 5 stones left, currentmax is updated to 2.
The first player can remove up to 3 stones (currentmax + 1), assuming 3 stones are removed and now there are 2 stones left, currentmax is updated to 3.
The second player can now simply remove the remaining two stones and win.

Now please play on: `ExpandingNim_instance`.

**Raw Score:**
Win: 1, Lose: 0

**Type:**
Competitive Two-player - Deterministic

Figure 8: Description of Expanding Nim.

---

**SharedNim**

IN A GAME CALLED SHARE CARD NIM, two players share the same sequence of cards. Once one player removes a card, the other can't use it. In each turn a player reveals a card and removes a number of stones equal to the number on the card. To win on a move, a player must play a card whose number is equal to the number of stones remaining. If your move is larger than the remaining stones, you lose.
Here is an example:
Initial number of stones: 10
Initial card list: [1, 2, 3, 4]
Player 1 places card 2, then the remaining stones is 8.
Player 2 places card 3, then the remaining stones is 5.
Player 1 places card 4, then the remaining stones is 1.
Player 2 places card 1, then the remaining stones is 0.
Player 2 wins.

Now please play on: `SharedNim_instance`.

**Raw Score:**
Win: 1, Lose: 0

**Type:**
Competitive Two-player - Deterministic

---

Figure 9: Description of Shared Nim.

---

**OptimalTouring**

Your task is to solve a puzzle named **Optimal Touring**. Each site has certain visiting hours. You have fixed a time you want to spend at each site which must all happen in one day. The time to go from site to site in minutes is the sum of street and avenue differences between them. On each day, you can start at any site you like. Your task is to visit as many sites as possible in one day.
The site data is site-data. What is the maximum value you can visit in one day?

Here is an example presenting the site-data:

```
sitesdata = {
1: {'avenue': 50, 'street': 96, 'desiredtime': 114, 'value': 3, '
    day': 1, 'beginhour': 6, 'endhour': 12},
2: {'avenue': 8, 'street': 23, 'desiredtime': 110, 'value': 186, '
    day': 1, 'beginhour': 9, 'endhour': 17},
3: {'avenue': 88, 'street': 69, 'desiredtime': 218, 'value': 3, '
    day': 1, 'beginhour': 9, 'endhour': 12},
4: {'avenue': 0, 'street': 95, 'desiredtime': 101, 'value': 86, '
    day': 1, 'beginhour': 6, 'endhour': 17},
5: {'avenue': 1, 'street': 48, 'desiredtime': 192, 'value': 199, '
    day': 1, 'beginhour': 5, 'endhour': 12}
}
```

Now please solve the problem: `OptimalTouring_instance`.

**Raw Score:**
The total value you get

**Type:**
Single-player - Deterministic

---

Figure 10: Description of Optimal Touring.

---

**CountMaximalCocktails**

Orphan diseases affect very few people, making the development of specific drugs challenging. To treat these diseases, a combination of drugs designed for other related conditions is often used. However, combining drugs can lead to harmful interactions. If no harmful interactions are present, combining the drugs may result in a synergistic effect, potentially benefiting the patient.

In this game, drugs are represented as nodes in a graph, and harmful interactions between drugs are represented as edges between nodes. The objective is to identify all maximal drug combinations, known as maximal cocktails, which correspond to the maximum independent sets in the graph. Players will explore how the addition of new interactions affects the number of maximal cocktails.

The current drug list is `nodes_list`, and the bad interaction list is `edges_list`. Each item in the interaction list is a tuple, and the two values in a tuple indicate that these two drugs have a bad interaction. what are the number of maximal cocktails?

For example, if the drug list is `[1, 2, 3, 4]` and the bad interaction list is `[(1, 2)]`, the maximal cocktails are `[1, 3, 4]` and `[2, 3, 4]`, so the number of maximal cocktails is 2.

**Raw Score:**
Success: 1, Failure: 0

**Type:**
Single-player - Deterministic

---

Figure 11: Description of Count Maximal Cocktails.

---

**MaxMaximalCocktails**

Orphan diseases affect very few people, making the development of specific drugs challenging. To treat these diseases, a combination of drugs designed for other related conditions is often used. However, combining drugs can lead to harmful interactions. If no harmful interactions are present, combining the drugs may result in a synergistic effect, potentially benefiting the patient.

In this game, drugs are represented as nodes in a graph, and harmful interactions between drugs are represented as edges between nodes. Based on the edges, we can identify all maximal drug combinations, known as maximal cocktails, which correspond to the maximum independent sets in the graph. And we can explore how the addition of new interactions affects the number of maximal cocktails.

Now, given a list nodes_list, each player plays in turn by adding one edge. The first player whose edge decreases the number of maximal cocktails loses. The edge should be in the format of (node1, node2), where node1 and node2 are two nodes in the list.

For example, if the list is `[1, 2, 3]`, and you are the first player, you can add the edge (1, 2), then the number of maximal cocktails is 2, which is larger than the number of maximal cocktails without the edge (1, 2), which is 1. So this addition is legal. But if your opponent adds the edge (2, 3) after you add the edge (1, 2), then the number of maximal cocktails is 3, which is also legal. After that, you will lose since you cannot add any edge to increase the number of maximal cocktails.

**Raw Score:**
Win: 1, Lose: 0

**Type:**
Competitive Two-player - Deterministic

---

Figure 12: Description of Max Maximal Cocktails.

**ExclusivityParticles**

There are some particles in a force field. By an exclusion principle, they must differ from one another by at least k in d dimensions, where each dimension is binary (for example, up or down spin). If it helps, think of the setting as a d-dimensional hypercube.

Now consider a two-player game. Suppose there are dimension dimensions, such that any two particles differ in at least distance dimensions. The two players take turns adding particles. The first player places a particle, and then the second player adds another, and so on. The game ends when a player cannot place a particle that satisfies the condition, and that player loses.

Please note that the way of computing the distance is the sum of the differences in each dimension. For example, the distance between `[0, 0]` and `[1, 1]` is 2. For instance, if the dimension is 3 and the required distance is 2, and you are the first player, you could place the first particle at `[0, 0, 0]`. The second player could then place the second particle at `[0, 1, 1]`. If you place the third particle at `[1, 0, 1]`, the second player cannot place a fourth particle that satisfies the condition and would lose.

**Raw Score:**
Win: 1, Lose: 0

**Type:**
Competitive Two-player - Deterministic

Figure 13: Description of Exclusivity Particles.

**ExclusivityProbes**

There are some number of particles in a force field. By an exclusion principle they must differ from one another by at least k among d dimensions where each dimension is a binary value (for example, up or down spin). If it helps, think of the setting as a d-dimensional hypercube.

Now suppose there are dimension dimensions and num_particles particles such that any two particles differ in at least distance dimensions. Each time, you can probe one position, and then I will respond 'yes' if a particle is at position p and 'no' otherwise. Your objective is to find all the positions of num_particles particles with as few probes as possible.

For example, if the dimension is 2, the number of particles is 2, and the distance is 1. We can probe the position `[0, 0]`, and if the response is 'yes', we only need one more probe to find the other particle because the particles can be either at locations `[0, 0]` and `[1, 1]` or at `[0, 1]` and `[1, 0]`. If the response is 'no', we need 3 more probes to find all the particles.

**Raw Score:**

- score < reference_score: 1
- score ≥ reference_score: reference_score / score

**Type:** Single-player - Stochastic

Figure 14: Description of Exclusivity Probes.

---

**RubyRisks**

You have three covered boxes of Burmese rubies before you. You know there are a total of [x] identical seven-carat rubies in the three boxes. You can ask for a certain number of rubies from each box. If you ask for more than there are, you get none from that box. Otherwise, you get what you asked for from that box. For now, suppose you must state your requests in advance for all three boxes and have no chance to change your mind; that is, with no feedback.

For example, you know that total rubies are 30.
In the first turn, you request 10 rubies.
Feedback: 10
In the second turn, you request 8 rubies.
Feedback: 8
In the third turn, you request 12 rubies.
Feedback: 0
Total rubies you get: 18

Now please guess the number of rubies: `RubyRisks_instance`.

**Raw Score:**
The final rubies you get from the game

**Type:**
Single-Player - Stochastic

---

Figure 15: Description of Ruby Risks.

---

**BeatOrBombDet**

As in many card games, particularly the game of War, each round involves each player choosing one card to play. Unlike other card games, each player can choose whether to compete with their card or to give it up. Points are calculated and accumulated after each round. At the end of the game, the player with the most points wins. A tie is possible, though unlikely. Now, let's go over the specific rules.

Rules:

- When the game starts, each player is given the same set of cards from 2 to A (with no Joker), one of each. The value of each card equals its numerical value, except for J, Q, K, and A, which are valued at 11, 12, 13, and 1, respectively.

- In each round, each player chooses and confirms one card from their set to play. They then decide whether to compete with this card or to give it up. This process is private, meaning each player will not see the decision made by their opponent. Once a decision is made, the card is removed from the player's set, whether it was played or given up.

- After both players have made their decisions, points are calculated as follows:
1. If both players choose to compete, the player with the higher-value card wins and is awarded points equal to their card value plus their opponent's card value.
2. If both players choose to give up, neither player receives any points.
3. If player A chooses to compete and player B chooses to give up, then player A is awarded points equal to their card value, while player B receives no points.

- After both players have played all their cards, the player with the most points is the winner.

**Raw Score:**
Win: 1, Lose: 0

**Type:**
Competitive Two-player - Deterministic

---

Figure 16: Description of Beat Or Bomb Det.

**BeatOrBombSto**

As in many card games, particularly the game of War, each round involves each player choosing one card to play. Unlike other card games, each player can choose whether to compete with their card or to give it up. Points are calculated and accumulated after each round. At the end of the game, the player with the most points wins. A tie is possible, though unlikely. Now, let's go over the specific rules.

Rules:

- At the start of the game, each player is given a set of 10 cards. Although the sets of cards may differ between players, the total value of the cards in each player's set is the same. The value of each card is equal to its numerical value, except for J, Q, K, and A, which have values of 11, 12, 13, and 1, respectively.

- In each round, each player chooses and confirms one card from their set to play. They then decide whether to compete with this card or to give it up. This process is private, meaning each player will not see the decision made by their opponent. Once a decision is made, the card is removed from the player's set, whether it was played or given up.

- After both players have made their decisions, points are calculated as follows:
1. If both players choose to compete, the player with the higher-value card wins and is awarded points equal to their card value plus their opponent's card value.
2. If both players choose to give up, neither player receives any points.
3. If player A chooses to compete and player B chooses to give up, then player A is awarded points equal to their card value, while player B receives no points.

- After both players have played all their cards, the player with the most points is the winner.

**Raw Score:**
Win: 1, Lose: 0

**Type:**
Competitive Two-player - Stochastic

Figure 17: Description of Beat Or Bomb Sto.

**MaxTarget**

You have 4 bags, each containing multiple coins with different values. Your goal is to maximize the total sum of coin values you collect by strategically choosing bags.

Before the game starts, you'll be informed of: 1. The coin values inside each bag 2. The total number of picks you can make

However, the actual order of the bags will be randomized. On each turn, you'll select a bag index, and a coin will be randomly drawn from that bag. For example, if you're told the bags contain `[1, 2]` and `[2, 3]`, but the actual order is `[[2, 3], [1, 2]]`, selecting bag index 0 will give you a random coin value from `[2, 3]`.

To maximize your score, you'll need to carefully consider the coin values in each bag and the number of remaining picks.

For example, if you're told the bags contain `[1, 2]` and `[3, 4]`, and the total number of picks is 2. If you pick bag 0 and get a coin value of 4, then in the next turn, you will know that bag 0 contains `[3, 4]` and bag 1 contains `[1, 2]`, and value 4 in bag 0 is removed and remaining values are `[3]`. So, if you pick bag 0 again, you will get a coin value of 3, which is bigger than the coin value of bag 1. So, you should pick bag 0 again to maximize your score.

Among the 4 bags, the coin values are random_bag[0], random_bag[1], random_bag[2], and random_bag[3]. You have max_guess picks in total. Please make your first pick.

**Raw Score:**

- score > reference_score: 1
- score ≤ reference_score: score / reference_score

**Type:**
Single-player - Stochastic

Figure 18: Description of Max Target.

**LargerTarget**

There are 4 bags, each containing multiple coins with different values. Two players take turns picking coins from a selection of bags. Your goal is to get a higher total sum of coin values than your opponent by strategically choosing bags.

Before the game starts, you'll be informed of:
1. The coin values inside each bag
2. The total number of picks you and your opponent can make

However, the actual order of the bags will be randomized. On each turn, you'll select a bag index, and a coin will be randomly drawn from that bag. For example, if you're told the bags contain [1, 2] and [2, 3], but the actual order is [[2, 3], [1, 2]], selecting bag index 0 will give you a random coin value from [2, 3].

To make your score higher than your opponent, you'll need to carefully consider the coin values in each bag and the number of remaining picks.

For example, if you're told the bags contain [1, 2] and [3, 4], and the total number of picks is 2. If your opponent pick bag 0 and get a coin value of 3, then in your turn, you will know that bag 0 contains [3, 4] and bag 1 contains [1, 2], and value 3 in bag 0 is removed and remaining values are [4]. So, if you pick bag 0 again, you will get a coin value of 4, which is bigger than the coin value of bag 1. So, you should pick bag 0 to make your score higher than your opponent.

Among the 4 bags, the coin values are random_bag[0], random_bag[1], random_bag[2], and random_bag[3]. And you and your opponent can make max_guess picks in total.

**Raw Score:**
Win: 1, Lose: 0

**Type:**
Competitive Two-player - Stochastic

Figure 19: Description of Larger Target.

## Takuzu

Your task is to solve a Takuzu puzzle. You are given a grid of size x * x.
The goal is to fill the grid with 0s and 1s following these rules:
1. Each row and column must contain an equal number of 0s and 1s.
2. No more than two identical numbers can be adjacent horizontally or vertically.
3. Each row and column must be unique.
4. There is only one valid solution.

For example, in a 4x4 grid: Input:

```
[[0, -1, 1, -1],
[-1, 0, -1, 1],
[1, -1, 0, -1],
[-1, 1, -1, 0]]
```

Output:

```
[[0, 1, 1, 0],
[1, 0, 0, 1],
[1, 0, 0, 1],
[0, 1, 1, 0]]
```

Now please solve this grid: `Takuzu_instance`.
**Raw Score:**
Success: 1, Failure: 0

**Type:**
Single-player - Deterministic

Figure 20: Description of Takuzu.

## K-Queens

the rules of the K-Queens puzzle are as follows:
- The player is given a grid of size NxN.
- The n queens are placed on the grid randomly at the beginning.
- The player must place N queens on the grid such that no two queens threaten each other.
- None of the queens share the same row, column, or diagonal.
Here is an example of the K-Queens puzzle:

```
[[ 0 0 0 0 0 1 0 0],
[0 0 0 0 0 0 0 1],
[0 0 0 0 1 0 0 0],
[0 0 0 0 0 0 1 0],
[0 1 0 0 0 0 0 0],
[0 0 1 0 0 0 0 0],
[0 0 0 0 0 0 0 1],
[0 1 0 0 0 0 0 0]]
```

To make it a KQueen, you can move

```
    [4, 1] to [4, 0]
```

Now please solve this grid: `K-Queens_instance`.
**Raw Score:**
Success: 1, Failure: 0

**Type:**
Single-player - Deterministic

Figure 21: Description of K-Queens.

**BidThat**

You and your opponent will bid on the items. You and your opponent are given specific dollars. The items are worth a list of prices: [price1, price2, price3].

You and your opponent will bid on the items. The highest bidder will get the items, and their current funds will be deducted by the highest bid. In the game, both you and your opponent bid without knowledge of the other's bid before result appears. Their current value will increase by the value of the items. If the bid is the same, the items will be passed. The game will end when all of the items are bought or passed. The player with the most current value will win the game.

An example of the game is as follows:
Your funds: 100
Opponent funds: 100
turn: 1
Items: [60, 50, 40]
You bid: 60
Opponent bid: 25
You win the item
You current funds: 100 - 60 = 40
You current value: 60
Opponent current funds: 100 Opponent current value: 0

Now please bid on: `BidThat_instance`.

**Raw Score:**
Win: 1, Lose: 0

**Type:**
Competitive Two-player - Deterministic

Figure 22: Description of Bid That.

1620
1621
1622
1623
1624
1625
1626
1627
1628
1629
1630
1631
1632
1633
1634
1635
1636
1637
1638
1639
1640
1641
1642
1643
1644
1645
1646
1647
1648
1649
1650
1651
1652
1653
1654
1655
1656
1657
1658
1659
1660
1661
1662
1663
1664
1665
1666
1667
1668
1669
1670
1671
1672
1673

**BidThatVikerey**

You and your opponent will bid on the items. You and your opponent are given specific dollars. The items are worth a list of prices: [price1, price2, price3].
You and your opponent will bid on the items. The highest bidder will get the items, and their current funds will be deducted by the second-highest bid. In the game, both you and your opponent bid without knowledge of the other's bid before result appears. Their current value will increase by the value of the items. If the bid is the same, the items will be passed. The game will end when all of the items are bought or passed. The player with the most current value will win the game.

An example of the game is as follows:
Your funds: 100
Opponent funds: 100
turn: 1
Items: [60, 50, 40]
You bid: 60
Opponent bid: 25
You win the item
You current funds: 100 - 25 = 75
You current value: 60
Opponent current funds: 100 Opponent current value: 0

Now please bid on: `BidThatVikerey_instance`.

**Raw Score:**
Success: 1, Failure: 0

**Type:**
Competitive Two-player - Deterministic

Figure 23: Description of Bid That Vikerey.

1674
1675
1676
1677
1678
1679
1680
1681
1682
1683
1684
1685
1686
1687
1688
1689
1690
1691
1692
1693
1694
1695
1696
1697
1698
1699
1700
1701
1702
1703
1704
1705
1706
1707
1708
1709
1710
1711
1712
1713
1714
1715
1716
1717
1718
1719
1720
1721
1722
1723
1724
1725
1726
1727

## Superply

This is a path-building board game played on a grid. The objective for Player 1 is to construct a path from the left side of the grid to the right, while Player 2 must build a path from the top to the bottom. A valid path is a sequence of adjacent same-value squares, where each square in the path must touch the next one either by a side or a corner.

During each turn, a player claims a square by selecting a grid position that satisfies the system-provided hint. If the chosen position is invalid, no changes are made, and the turn passes to the other player.

The hints are mathematical operations, such as "sum is less than 10," meaning that the sum of the numbers in the selected position must be less than 10 (row_index + column_index < 10). A player may choose any grid position that satisfies the given hint and is unoccupied.

The game board is a 6x6 grid, and it is 1-indexed. Initially, all grid values are set to 0. When a player correctly selects a grid position, the value of that position changes: 1 for Player 1, and 2 for Player 2. The first player to successfully build their path wins the game.

For example, if the hint is "product contains digit 6," and the grid is as follows:

```
[[0, 0, 0, 0, 0, 0],
[0, 0, 0, 0, 0, 0],
[0, 0, 0, 0, 0, 0],
[0, 0, 0, 0, 0, 0],
[0, 0, 0, 0, 0, 0],
[0, 0, 0, 0, 0, 0]]
```

If you are Player 1, you can select the position (1, 6), (6, 1), (2, 3), (3, 2) or (6, 6) because the product of the row and column indices is 6, 6, 6, 6 and 36, respectively, and they all contain the digit 6.

If you choose the position (6, 6), the grid becomes:

```
[[0, 0, 0, 0, 0, 0],
[0, 0, 0, 0, 0, 0],
[0, 0, 0, 0, 0, 0],
[0, 0, 0, 0, 0, 0],
[0, 0, 0, 0, 0, 0],
[0, 0, 0, 0, 0, 1]]
```

**Raw Score:** Win: 1, Lose: 0

**Type:** Competitive Two-player - Deterministic

Figure 24: Description of Superply.

# B EXPERIMENT SETUP

## B.1 LLMs CONFIGURATION

Table 7: Details of the LLMs evaluated in PUZZLEPLEX.

| Model | Creator | Version | Access Time | License | Input Modalities |
|---|---|---|---|---|---|
| **GPT-3.5-Turbo** | OpenAI | gpt-3.5-turbo-0125 | 2024.1 | Proprietary | text |
| **GPT-4o** | OpenAI | gpt-4o-0513 | 2024.5 | Proprietary | text & image |
| **Claude-3.5** | Anthropic | claude-3.5-sonnet-0620 | 2024.6 | Proprietary | text & image |
| **Qwen2** | Alibaba | Qwen2-72B-Instruct | 2024.1 | Open-source | text |
| **Mistral-8x7B** | MistralAI | Mistral-8x7B-Instruct-v0.2 | 2023.12 | Open-source | text |
| **Mixtral-8x22B** | MistralAI | Mixtral-8x22B-Instruct-v0.1 | 2024.4 | Open-source | text |
| **Pixtral** | MistralAI | pixtral-12b-240910 | 2024.9 | Open-source | text & image |
| **Gemini-1.5** | Google | Gemini-1.5-Pro | 2024.2 | Proprietary | text & image |
| | | Gemini-1.5-Flash | 2024.5 | Proprietary | text & image |
| **Llama-3** | Meta | Llama-3-8b-instruct | 2024.4 | Open-source | text |
| | | Llama-3-70B-Instruct | 2024.4 | Open-source | text |
| | | Llama-3.1-405B-Instruct | 2024.6 | Open-source | text |
| | | Llama-3.2-vision-90b | 2024.9 | Open-source | text & image |
| | | Llama-3.2-vision-11b | 2024.9 | Open-source | text & image |

## B.2 CUSTOMIZED MODEL CONFIGURATION

Table 8: Overview of puzzle games and their basic strategies. For text-image puzzles, we apply strategies similar to those used in corresponding text-only puzzles.

| Puzzle Name | Baseline Strategy | |
|---|---|---|
| | Easy | Intermediate |
| Sudoku (Wikipedia, 2024a) | SMT Solver | SMT Solver |
| SudoKill | Random | Greedy |
| Tidy Tower (Shasha, 2023) | Dynamic Programming | Dynamic Programming |
| Card Nim (Shasha, 2022a) | Random | Dynamic Programming |
| Expanding Nim | Random | Dynamic Programming |
| Share Card Nim | Random | Dynamic Programming |
| Optimal Touring | Simulated Annealing Algorithm | Simulated Annealing Algorithm |
| Count Maximal Cocktails (Shasha, 2022c) | Brute-force | Brute-force |
| Max Maximal Cocktails | Random | Brute-force |
| Exclusivity Particles (Shasha, 2022b) | Brute-force | Greedy |
| Exclusivity Probes | Random | Greedy |
| Ruby Risks (Shasha, 2017) | Monte-Carlo Tree Search | Monte-Carlo Tree Search |
| Beat Or Bomb Det. | Random | Random |
| Beat Or Bomb Sto. | Random | Greedy |
| Max Target | Greedy | Greedy |
| Larger Target | Random | Greedy |
| Takuzu (Wikipedia, 2024b) | Breadth-First Search | Breadth-First Search |
| KQueens | Breadth-First Search | Breadth-First Search |
| Bid That | Monte-Carlo Tree Search | Monte-Carlo Tree Search |
| Bit That Vickrey | Monte-Carlo Tree Search | Monte-Carlo Tree Search |
| Superply | Random | Searching |

### B.3 IMPLEMENTATION DETAILS OF MODEL INFERENCE

We use APIs to evaluate several models: GPT-4, GPT-3.5-turbo, Claude 3.5 Sonnet, Gemini 1.5 Pro, and Gemini 1.5 Flash. For other models, we utilize Hugging Face Transformers (Wolf et al., 2020) inference on $8 \times$ H100 and $8 \times$ A100.

### B.4 OPERATION EXTRACTION

For the raw output of LLMs, we use regular expressions to extract data. Each time we call an LLM, we allow up to 5 attempts. If the LLM cannot generate data in the requested format within these 5 attempts, we return None.

# C MORE RESULTS

## C.1 BRADLEY-TERRY MODEL STRENGTH

This section presents the **strength** (higher values indicate better performance) of each model in text puzzles and text-image puzzles. Additionally, two figures are provided for both text and text-image puzzles. The first figure is a heatmap where each cell represents the win probability of one model against another, while the second figure, also a heatmap, depicts the tie probabilities between models. The results show that, apart from customized methods, GPT-4o is the best model for text games, followed by Claude-3.5-sonnet as the second best, while Qwen2-72B stands out as the best open-sourced model. For text-image games, Claude-3.5-sonnet performs the best, with GPT-4o as the second best.

### C.1.1 TEXT GAMES

Table 9: Models ranked by strength in descending order in text games.

| Model | Strength |
|---|---|
| custom | 1.16 |
| gpt-4o | 0.52 |
| claude-3.5-sonnet | 0.35 |
| qwen2-72b | 0.23 |
| llama-3.1-70b | 0.06 |
| gemini-1.5-pro | -0.08 |
| llama-3.1-405b | -0.11 |
| mistral-8x22b | -0.12 |
| llama-3.1-8b | -0.43 |
| gemini-1.5-flash | -0.46 |
| gpt-3.5-turbo | -0.49 |
| mistral-8x7b | -0.62 |

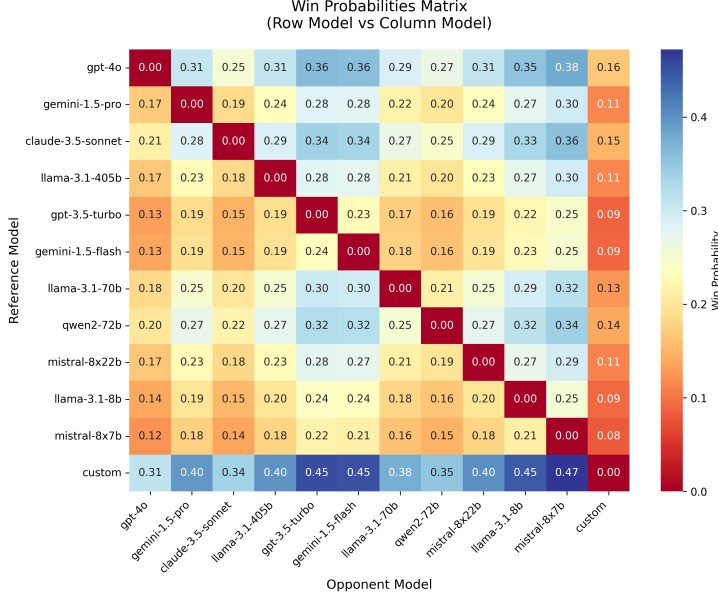

Figure 25: Win probabilities matrix in text games.

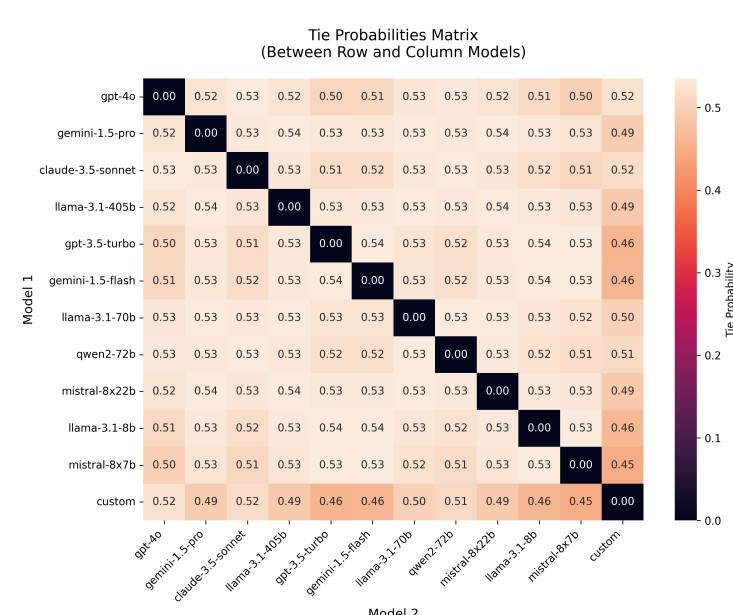

Figure 26: Tie probabilities matrix in text games.

### C.1.2 TEXT-IMAGE GAMES

Table 10: Models ranked by strength in descending order in text-image games.

| Model | Strength |
|---|---|
| custom | 5.68 |
| claude-3.5-sonnet | 0.05 |
| gpt-4o | -0.13 |
| gemini-1.5-flash | -0.57 |
| llama-3.2-vision-90b | -0.65 |
| llama-3.2-vision-11b | -1.10 |
| gemini-1.5-pro | -1.36 |
| pixtral | -1.91 |

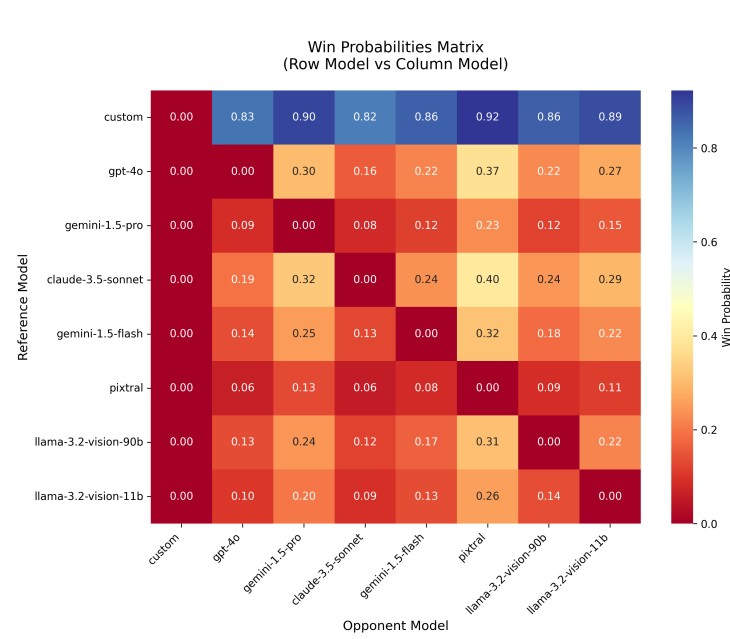

Figure 27: Win probabilities matrix in text-image games.

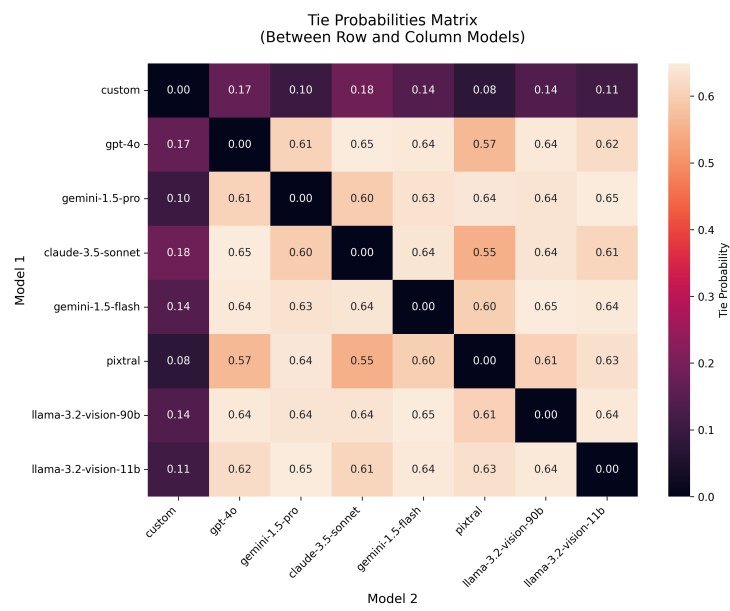

Figure 28: Tie probabilities matrix in text-image games.

## C.2 SCORE STATISTICS

To demonstrate the statistical significance of the results, tables presenting the mean, confidence intervals and standard deviations of model scores across different categories of puzzles are provided below.

Table 11: Statistics of text games.

| Model | Size | Single-player Deterministic | | | Single-player Stochastic | | | Competitive Deterministic | | | Competitive Stochastic | | | Total | | |
|---|---|---|---|---|---|---|---|---|---|---|---|---|---|---|---|---|
| | | Mean | 95% CI | SD | Mean | 95% CI | SD | Mean | 95% CI | SD | Mean | 95% CI | SD | Mean | 95% CI | SD |
| Baseline | - | 1.00 | (NaN, NaN) | 0.00 | 1.00 | (NaN, NaN) | 0.00 | 0.64 | (0.62, 0.66) | 0.48 | 0.62 | (0.60, 0.64) | 0.48 | 0.69 | (0.67, 0.70) | 0.46 |
| GPT-4o | - | 0.24 | (0.16, 0.32) | 0.43 | 0.59 | (0.47, 0.71) | 0.47 | 0.59 | (0.56, 0.61) | 0.49 | 0.45 | (0.29, 0.61) | 0.50 | 0.55 | (0.53, 0.58) | 0.50 |
| GPT-3.5-turbo | - | 0.13 | (0.07, 0.18) | 0.33 | 0.52 | (0.39, 0.64) | 0.49 | 0.36 | (0.33, 0.38) | 0.48 | 0.40 | (0.25, 0.55) | 0.49 | 0.34 | (0.32, 0.37) | 0.47 |
| Gemini-1.5-Pro | - | 0.21 | (0.14, 0.28) | 0.41 | 0.56 | (0.44, 0.69) | 0.48 | 0.43 | (0.40, 0.45) | 0.49 | 0.50 | (0.34, 0.66) | 0.50 | 0.42 | (0.39, 0.44) | 0.49 |
| Gemini-1.5-Flash | - | 0.18 | (0.11, 0.25) | 0.39 | 0.28 | (0.16, 0.39) | 0.44 | 0.46 | (0.43, 0.49) | 0.50 | 0.35 | (0.20, 0.50) | 0.48 | 0.43 | (0.40, 0.45) | 0.49 |
| Claude-3.5-Sonnet | - | 0.17 | (0.10, 0.23) | 0.37 | 0.59 | (0.47, 0.71) | 0.47 | 0.60 | (0.58, 0.63) | 0.49 | 0.50 | (0.34, 0.66) | 0.50 | 0.56 | (0.54, 0.59) | 0.50 |
| Llama-3.1 | 405B | 0.14 | (0.08, 0.20) | 0.35 | 0.42 | (0.30, 0.54) | 0.48 | 0.49 | (0.46, 0.52) | 0.50 | 0.35 | (0.20, 0.50) | 0.48 | 0.45 | (0.43, 0.48) | 0.50 |
| Llama-3.1 | 70B | 0.20 | (0.13, 0.27) | 0.40 | 0.53 | (0.49, 0.57) | 0.48 | 0.56 | (0.53, 0.59) | 0.50 | 0.39 | (0.34, 0.44) | 0.49 | 0.51 | (0.49, 0.53) | 0.50 |
| Llama-3.1 | 8B | 0.03 | (0.00, 0.05) | 0.16 | 0.60 | (0.56, 0.64) | 0.48 | 0.34 | (0.31, 0.36) | 0.47 | 0.26 | (0.22, 0.31) | 0.44 | 0.38 | (0.36, 0.40) | 0.48 |
| Mistral | 8×22B | 0.19 | (0.12, 0.26) | 0.39 | 0.62 | (0.58, 0.66) | 0.48 | 0.45 | (0.42, 0.48) | 0.50 | 0.28 | (0.24, 0.33) | 0.45 | 0.45 | (0.43, 0.47) | 0.50 |
| Mistral | 8×7B | 0.16 | (0.09, 0.22) | 0.37 | 0.55 | (0.52, 0.59) | 0.49 | 0.35 | (0.33, 0.38) | 0.48 | 0.26 | (0.22, 0.30) | 0.44 | 0.38 | (0.36, 0.40) | 0.48 |
| Qwen2 | 72B | 0.18 | (0.11, 0.24) | 0.38 | 0.64 | (0.60, 0.68) | 0.46 | 0.61 | (0.58, 0.64) | 0.49 | 0.41 | (0.36, 0.46) | 0.49 | 0.56 | (0.54, 0.58) | 0.49 |

Table 12: Statistics of text-image games.

| Model | Size | Single-player Deterministic | | | Competitive Deterministic | | | Total | | |
|---|---|---|---|---|---|---|---|---|---|---|
| | | Mean | 95% CI | SD | Mean | 95% CI | SD | Mean | 95% CI | SD |
| Baseline | - | 1.00 | (NaN, NaN) | 0.00 | 0.90 | (0.86, 0.94) | 0.30 | 0.91 | (0.87, 0.94) | 0.29 |
| GPT-4o | - | 0.00 | (NaN, NaN) | 0.00 | 0.56 | (0.49, 0.64) | 0.50 | 0.50 | (0.43, 0.57) | 0.50 |
| Gemini-1.5-Pro | - | 0.00 | (NaN, NaN) | 0.00 | 0.27 | (0.20, 0.34) | 0.44 | 0.24 | (0.18, 0.30) | 0.43 |
| Gemini-1.5-Flash | - | 0.00 | (NaN, NaN) | 0.00 | 0.46 | (0.38, 0.53) | 0.50 | 0.41 | (0.33, 0.48) | 0.49 |
| Claude-3.5-Sonnet | - | 0.00 | (NaN, NaN) | 0.00 | 0.61 | (0.53, 0.68) | 0.49 | 0.54 | (0.47, 0.61) | 0.50 |
| Pixtral | 12B | 0.00 | (NaN, NaN) | 0.00 | 0.14 | (0.08, 0.19) | 0.34 | 0.12 | (0.07, 0.17) | 0.33 |
| Llama-3.2 | 90B | 0.00 | (NaN, NaN) | 0.00 | 0.44 | (0.36, 0.51) | 0.50 | 0.39 | (0.32, 0.46) | 0.49 |
| Llama-3.2 | 11B | 0.00 | (NaN, NaN) | 0.00 | 0.33 | (0.26, 0.40) | 0.47 | 0.29 | (0.23, 0.36) | 0.46 |

## C.3 PLAY STATISTICS

Table 13: Statistics on the number of turns and tokens for all plays and legal plays of LLMs outputs in each game are presented, along with the percentage of legal plays out of the total plays for each puzzle.

| Name | Total Play | | | | | | | | Legal Play | | | | | | | | Legal Play Percentage |
|------|------------|--|--|--|--|--|--|--|------------|--|--|--|--|--|--|--|----------------------|
| | #Turns | | | | #Tokens | | | | #Turns | | | | #Tokens | | | | |
| | Min | Max | Mean | SD | Min | Max | Mean | SD | Min | Max | Mean | SD | Min | Max | Mean | SD | |
| Sudoku | 1 | 12 | 2.71 | 2.36 | 437 | 7973 | 1654.36 | 1253.31 | 10 | 12 | 10.88 | 0.60 | 2020 | 4564 | 3633.25 | 725.04 | 0.12 |
| SudoKill | 1 | 9 | 1.22 | 0.70 | 1303 | 6696 | 1715.01 | 481.67 | - | - | - | - | - | - | - | - | 0.00 |
| Tidy Tower | 1 | 20 | 5.95 | 7.65 | 589 | 31563 | 3308.04 | 5451.05 | 10 | 20 | 18.96 | 2.61 | 2423 | 31563 | 7933.54 | 6451.19 | 0.28 |
| Card Nim | 1 | 6 | 2.78 | 1.06 | 391 | 8473 | 994.17 | 462.18 | 1 | 5 | 2.91 | 0.81 | 424 | 8473 | 1076.61 | 573.82 | 0.28 |
| Expanding Nim | 1 | 11 | 3.76 | 2.01 | 525 | 4106 | 1246.39 | 571.33 | 2 | 11 | 4.65 | 1.54 | 738 | 4106 | 1435.97 | 541.11 | 0.54 |
| Share Card Nim | 1 | 4 | 2.26 | 0.95 | 381 | 4462 | 844.76 | 332.85 | 1 | 4 | 2.77 | 0.65 | 455 | 2503 | 959.85 | 304.78 | 0.55 |
| Optimal Touring | 1 | 2 | 1.01 | 0.11 | 1357 | 5214 | 2506.19 | 854.44 | 1 | 2 | 1.01 | 0.11 | 1357 | 5214 | 2509.86 | 855.81 | 0.75 |
| Count Maximal Cocktails | 1 | 1 | 1.00 | 0.00 | 349 | 2131 | 760.05 | 241.98 | 1 | 1 | 1.00 | 0.00 | 349 | 2131 | 765.59 | 242.61 | 0.96 |
| Max Maximal Cocktails | 1 | 3 | 1.60 | 0.68 | 470 | 3679 | 804.41 | 342.43 | 1 | 3 | 1.69 | 0.67 | 477 | 3290 | 775.17 | 309.70 | 0.76 |
| Exclusivity Particles | 1 | 9 | 3.88 | 1.79 | 420 | 8793 | 1116.90 | 593.49 | - | - | - | - | - | - | - | - | 0.00 |
| Exclusivity Probes | 1 | 130 | 23.10 | 18.59 | 384 | 65036 | 6295.74 | 6747.30 | 2 | 108 | 16.80 | 15.77 | 574 | 42581 | 3827.01 | 3983.23 | 0.49 |
| Ruby Risks | 1 | 3 | 2.90 | 0.32 | 405 | 9305 | 1054.63 | 443.21 | 3 | 3 | 3.00 | 0.00 | 672 | 9305 | 1070.12 | 445.07 | 0.91 |
| Beat Or Bomb Det. | 1 | 13 | 9.87 | 5.13 | 561 | 34260 | 2821.87 | 2696.80 | 13 | 13 | 13.00 | 0.00 | 1971 | 24181 | 3433.52 | 2530.30 | 0.69 |
| Beat Or Bomb Sto. | 1 | 10 | 8.86 | 2.83 | 636 | 23290 | 2383.73 | 1298.18 | 10 | 10 | 10.00 | 0.00 | 1641 | 20019 | 2593.16 | 1193.64 | 0.84 |
| Max Target | 1 | 15 | 8.38 | 3.11 | 524 | 13602 | 2270.03 | 1141.38 | 4 | 15 | 8.89 | 2.66 | 881 | 7505 | 2341.19 | 1032.72 | 0.90 |
| Larger Target | 1 | 11 | 8.05 | 1.60 | 622 | 28547 | 2948.03 | 1422.29 | 7 | 11 | 8.47 | 1.13 | 1558 | 28547 | 3023.99 | 1460.38 | 0.73 |
| Takuzu | 1 | 10 | 2.39 | 1.89 | 458 | 10324 | 2352.28 | 1751.12 | - | - | - | - | - | - | - | - | 0.00 |
| KQueens | 1 | 1 | 1.00 | 0.00 | 812 | 4663 | 1605.29 | 884.89 | - | - | - | - | - | - | - | - | 0.00 |
| Bid That | 1 | 3 | 2.79 | 0.55 | 325 | 2205 | 988.49 | 385.31 | 3 | 3 | 3.00 | 0.00 | 532 | 2103 | 1061.49 | 362.86 | 0.77 |
| Bid That Vickrey | 1 | 3 | 2.99 | 0.13 | 353 | 1787 | 947.11 | 302.52 | 3 | 3 | 3.00 | 0.00 | 422 | 1787 | 950.78 | 301.64 | 0.98 |
| Superply | 1 | 30 | 13.76 | 5.89 | 866 | 36759 | 6189.31 | 5870.22 | 5 | 30 | 14.14 | 6.40 | 1754 | 31228 | 5747.43 | 4167.54 | 0.72 |

## C.4 RESULTS OF TEXT-IMAGE GAME

Table 14: Results for the single-player scenario in the SudokuM puzzle (a version of the puzzle where the LLM is shown the Sudoku matrix image) and the same models also tested on a text version of Sudoku. The LLMs all did badly on the image data. Claude-3.5-Sonnet could solve half of the easy text-based Sudoku puzzles.

| Model | Size | SudokuM | | | | Sudoku | | | |
| | | Easy | | Inter. | | Easy | | Inter. | |
| | | Score | FIR | Score | FIR | Score | FIR | Score | FIR |
|---|---|---|---|---|---|---|---|---|---|
| Baseline | - | 1.00 | 0.00 | 1.00 | 0.00 | 1.00 | 0.00 | 1.00 | 0.00 |
| GPT-4o | - | 0.00 | 1.00 | 0.00 | 1.00 | 0.10 | 0.90 | 0.00 | 1.00 |
| Geminni-1.5-Pro | - | 0.00 | 1.00 | 0.00 | 1.00 | 0.00 | 1.00 | 0.00 | 1.00 |
| Geminni-1.5-Flash | - | 0.00 | 1.00 | 0.00 | 1.00 | 0.00 | 1.00 | 0.00 | 1.00 |
| Claude-3.5-Sonnet | - | 0.00 | 1.00 | 0.00 | 1.00 | **0.50** | **0.50** | 0.00 | 1.00 |
| Pixtral | 12B | 0.00 | 1.00 | 0.00 | 1.00 | – | – | – | – |
| Llama-3.2 | 90B | 0.00 | 1.00 | 0.00 | 1.00 | – | – | – | – |
| Llama-3.2 | 11B | 0.00 | 1.00 | 0.00 | 1.00 | – | – | – | – |

Table 15: Results for the multi-turn competitive two-player scenario for SudoKill (an competitive two-player version of Sudoku in which players alternate placing numbers in empty grid cells until one violates the Sudoku rules) and Superply (Superply is an competitive two-player game played on a multiplication table with both blanks and numbers). In the SudoKill and Superply cases, the data is represented textually. In the SudoKillM and SuperplyM cases, the data is represented visually as a matrix. The adversaries are the LLMs shown in the Model column and a custom opponent (backtracking and searching for Sudokill and greedy for Superply). Claude-3.5-Sonnet performed best at the (visual) SuperplyM variants. GPT-4o performed best at easy Superply.

| Model | SudoKillM | | | | SudoKill | | | | SuperplyM | | | | Superply | | | |
| | Easy | | Inter. | | Easy | | Inter. | | Easy | | Inter. | | Easy | | Inter. | |
| | Score | FIR | Score | FIR | Score | FIR | Score | FIR | Score | FIR | Score | FIR | Score | FIR | Score | FIR |
|---|---|---|---|---|---|---|---|---|---|---|---|---|---|---|---|---|
| GPT-4o | 0.00 | 1.00 | 0.00 | 1.00 | 0.00 | 1.00 | 0.00 | 1.00 | **0.50** | 0.00 | 0.20 | 0.00 | **0.70** | 0.00 | 0.20 | 0.00 |
| Gemini-1.5-Pro | 0.00 | 1.00 | 0.00 | 1.00 | 0.00 | 1.00 | 0.00 | 1.00 | 0.20 | 0.70 | 0.00 | 0.90 | 0.20 | 0.30 | 0.10 | 0.20 |
| Gemini-1.5-Flash | 0.00 | 1.00 | **0.10** | **0.90** | 0.00 | 1.00 | 0.00 | 1.00 | 0.40 | 0.10 | 0.00 | 0.00 | 0.10 | 0.00 | 0.00 | 0.00 |
| Claude-3.5-Sonnet | 0.00 | 1.00 | 0.00 | 1.00 | **0.10** | **0.90** | 0.00 | 1.00 | **0.50** | 0.00 | **0.40** | 0.00 | 0.40 | 0.10 | **0.30** | 0.00 |
| Pixtral-12B | 0.00 | 1.00 | 0.00 | 1.00 | – | – | – | – | 0.00 | 1.00 | 0.00 | 1.00 | – | – | – | – |
| Llama-3.2-90B-Vision | 0.00 | 1.00 | 0.00 | 1.00 | – | – | – | – | 0.40 | 0.00 | 0.00 | 0.00 | – | – | – | – |
| Llama-3.2-11B-Vision | 0.00 | 1.00 | 0.00 | 1.00 | – | – | – | – | 0.10 | 0.10 | 0.00 | 0.00 | – | – | – | – |

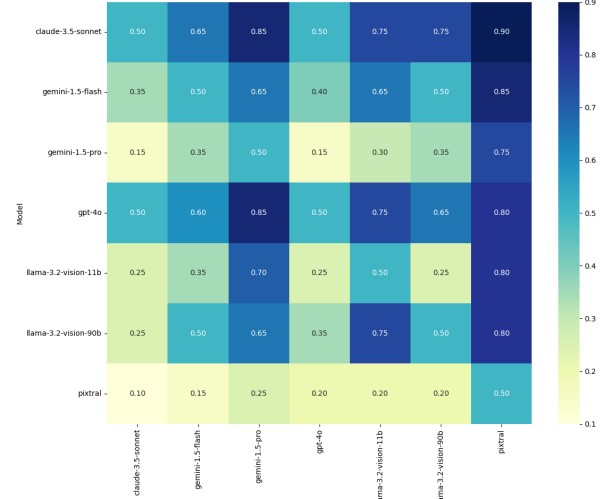

| Model | Win Rate |
|---|---|
| Claude-3.5-sonnet | 0.73 |
| GPT-4o | 0.69 |
| Gemini-1.5-flash | 0.57 |
| Llama-3.2-vision-90b | 0.55 |
| Llama-3.2-vision-11b | 0.43 |
| Gemini-1.5-pro | 0.34 |
| Pixtral | 0.18 |

Figure 29: The figure on the left compares the results of competitive two-player multi-turn deterministic text-image games between pairs of models. The table on the right demonstrates the **Win Rate** of models ranked in descending order. Claude-3.5-Sonnet performs the best with GPT-4o a close second.

# D    ERROR ANALYSIS

## D.1    ERROR EXPLANATION

Table 16: Error type definitions in PUZZLEPLEX.

| Error Type & Explanation |
|---|
| **Reasoning and Planning Error** (63%): This type of error occurs in two main forms: an LLM makes illegal moves or it makes legal moves, but fails to develop and execute effective strategies. This reflects both a failure to determine legal moves and a failure to connect valid moves with their strategic implications, resulting in suboptimal decision-making. The key distinction is that these errors occur at the reasoning level, whether in move validation or in strategic planning. |
| **Comprehension Error** (12%): This error represents a fundamental misunderstanding or incomplete grasp of the game's core rules leading to planning and reasoning that may be logically sound suboptimal because it's based on misunderstood premises. |
| **Memorization Error** (11%): This error type represents the LLM's failure to maintain and utilize historical game state information over multiple turns leading to repetitive or contradictory actions. |
| **Perception Error** (7%): This error occurs at the input processing level, where the LLM fails to accurately interpret or extract the current game state in visual or multimodal contexts. |
| **Other Error** (7%): This category includes all remaining errors. |

## D.2    REASONING AND PLANNING ERRORS

**System**
Your task it to solve a puzzle named 'Count Maximal Cocktails'. Orphan diseases affect very few people, making the development of specific drugs challenging. To treat these diseases, a combination of drugs designed for other related conditions is often used. However, combining drugs can lead to harmful interactions. If no harmful interactions are present, combining the drugs may result in a synergistic effect, potentially benefiting the patient. In this game, drugs are represented as nodes in a graph, and harmful interactions between drugs are represented as edges between nodes. The objective is to identify all maximal drug combinations, known as maximal cocktails, which correspond to the maximum independent sets in the graph. Players will explore how the addition of new interactions affects the number of maximal cocktails. The current drug list is [1, 2, 3, 4, 5], and the bad interaction list is [(1, 3), (2, 3), (4, 5), (1, 5)]. Each item in the interaction list is a tuple, and the two values in a tuple indicate that these two drugs have a bad interaction. what are the number of maximal cocktails? For example, if the drug list is [1, 2, 3, 4] and the bad interaction list is [(1, 2)], the maximal cocktails are [1, 3, 4] and [2, 3, 4], so the number of maximal cocktails is 2. Provide the output in the following format: Reasoning: ... Operation: Output the maximal cocktails in the format `maximal_cocktails = int`.

**GPT-4o**
Reasoning: To find the maximal cocktails, we need to determine the maximal independent sets in the graph represented by the drugs and their interactions. An independent set is a set of nodes such that no two nodes in the set are adjacent. A maximal independent set is one that cannot be enlarged by adding more nodes without violating the independence condition. Let's consider the interactions: - Drug list: [1, 2, 3, 4, 5] - Bad interactions: [(1, 3), (2, 3), (4, 5), (1, 5)] First, we visualize the graph: ``` 1 -- 3 2 -- 3 4 -- 5 1 -- 5 ``` Now, we identify all the maximal independent sets: 1. Consider each drug and combine with others while avoiding bad interactions. 2. Possible maximal cocktails: - Start with drug 1: Cannot pair with 3 or 5. - Possible set: [1, 2, 4] - Start with drug 2: Cannot pair with 3. - Possible sets: [2, 4, 5], [2, 4, 1] (2 can pair with 1, 4, and 5) - Start with drug 3: Cannot pair with 1 or 2. - Possible sets: [3, 4, 5] (3 can pair with 4 and 5, avoiding others) - Start with drug 4: Cannot pair with 5. - Possible sets: [4, 1, 2], [4, 2, 3] - Start with drug 5: Cannot pair with 1 or 4. - Possible set: [5, 2, 3] Ensuring each set is maximal: - [2, 4, 5] - [3, 4, 5] - [1, 2, 4] - [5, 2, 3] Operation: Output the maximal cocktails in the format `maximal_cocktails = 4`.

Figure 30: An example of a reasoning error by GPT-4o in the puzzle *Count Maximal Cocktails* occurs during the counting of maximal cocktails. When evaluating a node, the model considers only the bad interactions directly associated with that node and fails to account for bad interactions within the set itself. For instance, in the context of drug 2, the model overlooks the fact that 4 and 5 also have a bad interaction in the possible set [2, 4, 5].

| | |
|---|---|
| System | You need to play a puzzle named 'Sudokill' against another player. ... The initial grid is [[0, 0, 3, 1, 6, 7, 0, 0, 2], [7, 0, 1, 8, 4, 9, 0, 3, 5], [9, 0, 8, 0, 5, 0, 1, 7, 4], [0, 0, 0, 0, 1, 0, 0, 8, 7], [0, 0, 0, 4, 0, 8, 2, 0, 0], [0, 0, 6, 7, 0, 2, 4, 0, 3], [0, 3, 9, 5, 8, 0, 0, 0, 1], [8, 7, 4, 0, 0, 0, 0, 6, 9], [6, 1, 5, 9, 7, 4, 0, 2, 0]]. Here, you are the first player. You can place a number in any unoccupied space. |
| Llama | ... fill cell (3, 3) with the value 6. |
| System | ... Your opponent fill the cell at (3, 3) with the value 6. Now the grid becomes [[0, 0, 3, 1, 6, 7, 0, 0, 2], [7, 0, 1, 8, 4, 9, 0, 3, 5], [9, 0, 8, 0, 5, 0, 1, 7, 4], [0, 0, 0, 6, 1, 0, 0, 8, 7], [0, 0, 0, 4, 0, 8, 2, 0, 0], [0, 0, 6, 7, 0, 2, 4, 0, 3], [0, 3, 9, 5, 8, 0, 0, 0, 1], [8, 7, 4, 0, 0, 0, 0, 6, 9], [6, 1, 5, 9, 7, 4, 0, 2, 0]]. Now it is your turn. |
| GPT-4o | To decide my move, I need to place a number in either the same row (3) or column (3) where the opponent placed their last number, ensuring I don't violate any Sudoku rules. I'll aim to restrict my opponent's next moves by blocking possible avenues for them. First, I'll analyze the available positions in row 3 and column 3: - Row 3 possibilities after the opponent's move: (3, 0), (3, 1), (3, 2), (3, 5), (3, 6) - Column 3 possibilities after the opponent's move: (5, 3), (7, 3) From these possibilities, I'll choose a cell that allows me to place a number such that there's less flexibility for the opponent. Let's examine the options: ### Choice of Cell and Value 1. **Cell (3, 0):** Only values 5, 9 can fit. 2. **Cell (3, 1):** Values 2, 5, 9 can fit. 3. **Cell (3, 2):** Values 2, 5 can fit. 4. **Cell (3, 5):** Values 3, 5 can fit. 5. **Cell (3, 6):** Values 3, 5, 9 can fit. 6. **Cell (5, 3):** Values 3, 9 can fit. 7. **Cell (7, 3):** Values 3, 9 can fit. By filling **(3, 5)** with **5**, the opponent will have limited options and may end up in a situation with fewer legal moves, depending on their next steps. ### Conclusion Therefore, placing '5' at position (3, 5) constrains the opponent's choices significantly for their upcoming move. Reasoning: By filling cell (3, 5) with 5, I'm reducing the available options for my opponent, forcing them into a narrower set of choices that may limit their strategic flexibility and increase the chance of them making a mistake. Operation: `operation = [(3, 5), 5]` |
| System | ... Your opponent fill the cell at (3, 5) with the value 5. Now the grid becomes [[0, 0, 3, 1, 6, 7, 0, 0, 2], [7, 0, 1, 8, 4, 9, 0, 3, 5], [9, 0, 8, 0, 5, 0, 1, 7, 4], [0, 0, 0, 6, 1, 5, 0, 8, 7], [0, 0, 0, 4, 0, 8, 2, 0, 0], [0, 0, 6, 7, 0, 2, 4, 0, 3], [0, 3, 9, 5, 8, 0, 0, 0, 1], [8, 7, 4, 0, 0, 0, 0, 6, 9], [6, 1, 5, 9, 7, 4, 0, 2, 0]]. Now it is your turn. |
| Llama | ... fill cell (5, 4) with the value 3. |

Figure 31: An example of flawed reasoning by GPT-4o in the game SudoKill while playing against Llama-3.1-405B, despite GPT-4o ultimately winning the game. The text highlighted in red indicates that certain cells are not eligible to be filled, and for some candidate cells, specific values are also ineligible to be filled.

Step 1:
[[0, 0, 0, 0], [0, 0, 0, 0], [3, 4, 0, 2], [2, 1, 4, 0]]
Step 2:
[[1, 0, 0, 0], [0, 0, 0, 0], [3, 4, 0, 2], [2, 1, 4, 0]]
Step 3:
[[1, 0, 0, 2], [0, 0, 0, 0], [3, 4, 0, 2], [2, 1, 4, 0]]

GPT-4o

Step 1:
[[0, 0, 0, 0], [0, 0, 0, 0], [3, 4, 0, 2], [2, 1, 4, 0]]
Step 2:
[[0, 0, 0, 0], [0, 0, 0, 0], [3, 4, 1, 2], [2, 1, 4, 0]]
Step 3:
[[0, 0, 0, 0], [0, 0, 0, 0], [3, 4, 1, 2], [2, 1, 4, 3]]
Step 4:
[[4, 0, 0, 0], [0, 0, 0, 0], [3, 4, 1, 2], [2, 1, 4, 3]]
Step 5:
[[4, 0, 0, 0], [1, 0, 0, 0], [3, 4, 1, 2], [2, 1, 4, 3]]
Step 6:
[[4, 3, 0, 0], [1, 0, 0, 0], [3, 4, 1, 2], [2, 1, 4, 3]]
Step 7:
[[4, 3, 0, 0], [1, 2, 0, 0], [3, 4, 1, 2], [2, 1, 4, 3]]
Step 8:
[[4, 3, 2, 0], [1, 2, 0, 0], [3, 4, 1, 2], [2, 1, 4, 3]]
Step 9:
[[4, 3, 2, 1], [1, 2, 0, 0], [3, 4, 1, 2], [2, 1, 4, 3]]
Step 10:
[[4, 3, 2, 1], [1, 2, 3, 0], [3, 4, 1, 2], [2, 1, 4, 3]]
Step 11:
[[4, 3, 2, 1], [1, 2, 3, 4], [3, 4, 1, 2], [2, 1, 4, 3]]

Claude-3.5-Sonnet

Figure 32: An example comparing each step of cell value placement between GPT-4o and Claude-3.5-Sonnet in the game *Sudoku*. The values highlighted in red indicate the position and value to be filled in at each step. It shows that Claude-3.5-Sonnet can identify a better solving path.

### D.3 COMPREHENSION ERRORS

| | |
|---|---|
| System | You have 4 bags, each containing multiple coins with different values. Your goal is to maximize the total sum of coin values you collect by strategically choosing bags. Before the game starts, you'll be informed of: 1. The coin values inside each bag 2. The total number of picks you can make. However, the actual order of the bags will be randomized.
…
Among the 4 bags, the coin values are [4, 4, 4, 2], [1, 4, 3, 1], [2, 3, 3], and [3, 1]. You have 6 picks in total. Please make your first pick. |
| GPT-4o | … Choose bag 0 to try and collect a high-value coin first. bag_index = [0] |
| System | Your last guess was 0 and you received a coin value of 2. The total value you have received so far is 2. You have 5 picks left. Please make your next pick. |
| GPT-4o | … Continue to pick from Bag 0 for the high potential coin value. bag_index = [0] |
| System | Your last guess was 0 and you received a coin value of 3. The total value you have received so far is 5. You have 4 picks left. Please make your next pick. |
| GPT-4o | Reasoning: Focusing on the new status of the bags: 1. Bag 0: [4, 4] 2. Bag 1: [1, 4, 3, 1] → Average = 2.25 3. Bag 2: [2, 3, 3] → Average = 2.67 4. Bag 3: [3, 1] → Average = 2 The remaining average value for Bag 0 is still high at 4. This is still the best option because it has two 4s left. Operation: Continue picking from Bag 0. bag_index = [0] |

| | |
|---|---|
| System | You have 4 bags, each containing multiple coins with different values. Your goal is to maximize the total sum of coin values you collect by strategically choosing bags. Before the game starts, you'll be informed of: 1. The coin values inside each bag 2. The total number of picks you can make. However, the actual order of the bags will be randomized.
…
Among the 4 bags, the coin values are [4, 3, 3, 4], [4, 2, 2, 1], [3, 3, 2], and [1, 1]. You have 8 picks in total. Please make your first pick. |
| Llama | … bag_index = [0] |
| System | Your last guess was 0 and you received a coin value of 1. The total value you have received so far is 1. You have 7 picks left. Please make your next pick. |
| Llama | With the new information, we can update our knowledge about the bags. Since we picked from bag 0 and received a coin value of 1, but the original values in bag 0 were [4, 3, 3, 4], we can infer that the actual values in bag 0 are likely not what we were initially told. However, this information is not very helpful at this point. Given that we still don't know the actual order of the bags, our strategy remains the same: we'll choose the bag with the next highest average coin value. The average coin values for the bags are still: Bag 0: unknown (but likely not 3.5) Bag 1: 2.25 Bag 2: 2.67 Bag 3: 1 Reasoning: Since bag 0 is uncertain, we'll choose the bag with the next highest average coin value, which is bag 2. Operation: bag_index = [2] |

Figure 33: An example of a comprehension error by GPT-4o in the puzzle *Max Target* can be observed when comparing it to Llama-3.1-405B. GPT-4o fails to thoroughly understand the rules, particularly that the provided bags are randomized, whereas Llama-3.1-405B correctly accounts for this aspect.

### D.4 MEMORIZATION ERRORS

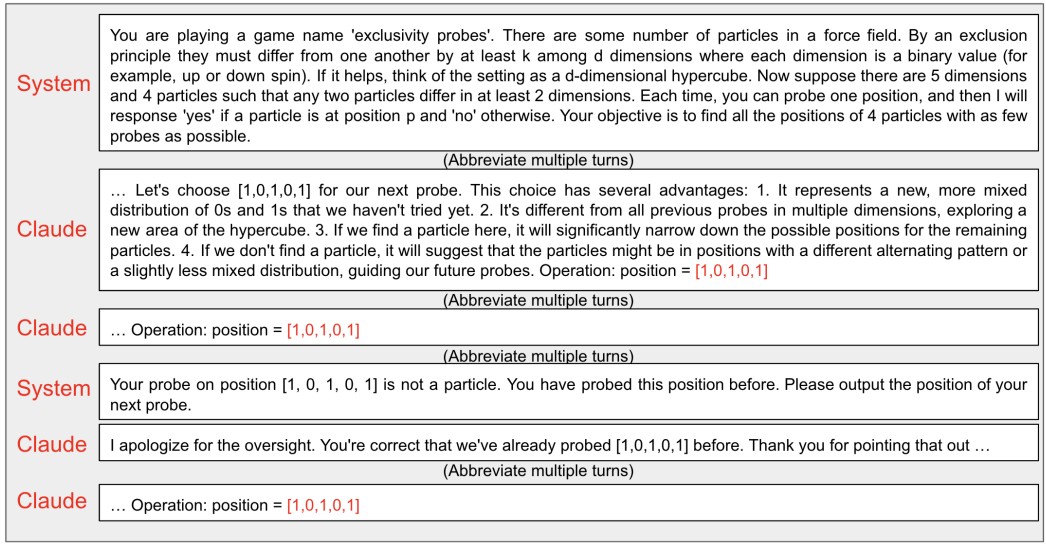

| | |
|---|---|
| System | You are playing a game name 'exclusivity probes'. There are some number of particles in a force field. By an exclusion principle they must differ from one another by at least k among d dimensions where each dimension is a binary value (for example, up or down spin). If it helps, think of the setting as a d-dimensional hypercube. Now suppose there are 5 dimensions and 4 particles such that any two particles differ in at least 2 dimensions. Each time, you can probe one position, and then I will response 'yes' if a particle is at position p and 'no' otherwise. Your objective is to find all the positions of 4 particles with as few probes as possible. |
| | (Abbreviate multiple turns) |
| Claude | … Let's choose [1,0,1,0,1] for our next probe. This choice has several advantages: 1. It represents a new, more mixed distribution of 0s and 1s that we haven't tried yet. 2. It's different from all previous probes in multiple dimensions, exploring a new area of the hypercube. 3. If we find a particle here, it will significantly narrow down the possible positions for the remaining particles. 4. If we don't find a particle, it will suggest that the particles might be in positions with a different alternating pattern or a slightly less mixed distribution, guiding our future probes. Operation: position = [1,0,1,0,1] |
| | (Abbreviate multiple turns) |
| Claude | … Operation: position = [1,0,1,0,1] |
| | (Abbreviate multiple turns) |
| System | Your probe on position [1, 0, 1, 0, 1] is not a particle. You have probed this position before. Please output the position of your next probe. |
| Claude | I apologize for the oversight. You're correct that we've already probed [1,0,1,0,1] before. Thank you for pointing that out … |
| | (Abbreviate multiple turns) |
| Claude | … Operation: position = [1,0,1,0,1] |

Figure 34: An example of a memorization error by Claude-3.5-Sonnet in the game Exclusivity Probes demonstrates that the model repeatedly revisited positions it had already visited. Eventually, it reached the maximum turn limit of 64 and terminated.

## D.5 OTHER ERRORS

| System | You are required to play a game called Superply with another player. This is a path-building board game played on a grid. The objective for Player 1 is to construct a path from the left side of the grid to the right, while Player 2 must build a path from the top to the bottom. A valid path is a sequence of adjacent same-value squares, where each square in the path must touch the next one either by a side or a corner … Now the grid becomes [[1, 0, 0, 0, 0, 0], [0, 0, 0, 0, 0, 0], [0, 0, 0, 0, 0, 0], [0, 0, 0, 0, 0, 0], [0, 0, 0, 0, 0, 0], [0, 0, 0, 0, 0, 0]]. The current hint is: product is greater than 19. Now it is your turn. |
|---|---|
| Gemini | Reasoning: I need to find a grid position where the product of the row index and column index is greater than 19. The position (4, 5) satisfies this condition (4 * 5 = 20). Operation: `operation = (4, 5)` |
| | (Abbreviate multiple turns) |
| System | Your opponent has chosen the position (5, 6). This position is valid for the hint. Now the grid becomes [[1, 1, 1, 2, 0, 1], [0, 0, 0, 2, 1, 0], [0, 0, 0, 0, 0, 0], [0, 0, 0, 2, 2, 0], [0, 0, 0, 2, 1, 1], [0, 0, 0, 0, 0, 0]]. The current hint is: product is even. Now it is your turn. |
| Gemini | Reasoning: I need to find a grid position where the product of the row and column indices is even. Given my current path and the hint, position (2, 5) is a strategic choice as it allows me to continue building downwards and has an even product (2 * 5 = 10). Operation: `operation = (2, 5)` |

Figure 35: An example of an error by Gemini-1.5-Pro occurs in the puzzle Superply, where the model sometimes prioritizes numerical reasoning and overlooks the fact that a position is already filled.

