# OpenReview forum: "PuzzlePlex: A Benchmark to Evaluate the Reasoning and Planning of Large Language Models on Puzzles"
_ICLR.cc/2025/Conference — Submitted to ICLR 2025_

### Official Review · Reviewer_X2oB · 2024-11-01

**Soundness:** 3
**Presentation:** 2
**Contribution:** 3
**Rating:** 6
**Confidence:** 4

**Summary:**

This paper provides a new benchmark called PuzzlePlex for evaluating reasoning and planning in LLMs. The benchmark consists of 24 puzzles that can be categorized into various types: single-player determinstic vs. stochastic games, and adversarial deterministic vs. stochastic games. In addition, the puzzles are parametrizable in order to provide multiple levels of difficulty. The benchmark is the first to include both single-player and adversarial (2-player) scenarios, as well as to include text-image puzzles. The work evaluates a suite of models on the benchmark and finds that while GPT-4o is generally the strongest model, there is still significant room for improvement.

**Strengths:**

The paper has several strengths:
1. The work provides a timely benchmark: there is currently a lot of interest in the field in making LLMs better planners an reasoners, and evaluation is an important component of this.
2. Included in the benchmark are classic game-playing and puzzle-solving baseline techniques against which to compare LLMs.
3. The benchmark provides multiple levels of difficulty in order to retain its relevance and usefulness as LLMs grow stronger over time.
4. The benchmark provides a good variety of game properties: single-player vs. adversarial (two-player), deterministic vs. stochastic., and text vs. text-image.

**Weaknesses:**

The paper has several weaknesses:
1. There are no CIs or SEs included in the result tables, making it hard to judge sometimes whether results are significant. It would be great if these statistics could be added (probably best in the appendix).
2. It doesn’t seem the authors tried any agent baselines like ReAct [1], Reflexion [2], or inference time methods like Tree-of-Thoughts (ToT) [3]. These kind of methods have been shown to dramatically increase performance sometimes (e.g. Game of 24 in ToT paper which went from 4% success rate to 74% success rate). Without these, it’s unclear what the real ceiling is for these models currently on this benchmark, making it hard to assess how challenging and useful the benchmark will be going forward.
3. The paper generally feels a bit rushed and could use a bit more structure. For example, there is quite a few result tables but oftentimes the captions don’t provide short takeaways (e.g. see the caption for Table 6) making it hard to remember what the purpose of each result table is. In addition, when following the Github link for the code it turns out that some of the figures in the paper are incorrect and the correct ones are included in the repo. These kind of things indicate the paper could potentially benefit from another round of refinement.
4. In section 4.5, the authors identify several types of errors that GPT-4o and Qwen2-72B encounter when playing the puzzles. However, there are no quantitative examples illustrating these kind of errors, nor any quantitative metrics as to how often each of these error types happen, which one is the most common, etc.

[1] Yao, Shunyu, et al. "React: Synergizing reasoning and acting in language models." arXiv preprint arXiv:2210.03629 (2022).

[2] Shinn, Noah, et al. "Reflexion: Language agents with verbal reinforcement learning." Advances in Neural Information Processing Systems 36 (2024).

[3] Yao, Shunyu, et al. "Tree of thoughts: Deliberate problem solving with large language models." Advances in Neural Information Processing Systems 36 (2024).

**Questions:**

1. Figure 1 includes an “Evaluator” which is discussed a bit further in Section 3.1 but for the rest doesn’t appear in the paper. Could the authors elaborate a bit on what kind of system the evaluator is and how exactly it works?
2. The paper uses the term “adversarial game” quite a bit. What about these games is adversarial? I don’t fully understand the difference between this and standard competitive 2-player games. Could the authors please clarify?
3. For the tables on the right in Figures 2 & 3, who is the win rate computed against?
4. “To mitigate the risk of exceeding the contextual length, given the likelihood of multiple turns in our games, our evaluation primarily adopts a zero-shot CoT approach.” Do the authors have a sense of how long the current context windows are getting? Models these days can handle pretty large context lengths, so I would be very curious if you’re running into those limits.
5. In Section 4.1 under the paragraph “Adversarial Text Puzzles”, should Table 4 be Table 3 instead?

**Details Of Ethics Concerns:**

No concerns.

---

> ### Author Response · Authors · 2024-11-24
>
> Thank you for your thoughtful feedback. I'm pleased to address each point below.
>
> ## Weaknesses
> ### W1. Lack of CIs or SEs
> Thank you for your suggestion. We have added confidence intervals and standard errors for the results in Appendix C.2 in the revised version. This addition should make it easier to judge the statistical significance of the results.
>
> ### W2. Lack of Agent Baselines and Advanced Prompting Strategies
> We have included experiments regarding Tree-of-Thought (ToT) prompting and one-shot prompting in Section 4.4 of the revised paper. These techniques demonstrated improvements for smaller sizes of each puzzle but showed minimal benefit for larger sizes of those same puzzles. This demonstrates the needs for future work to develop better prompting strategies for larger scale puzzles.
>
> Regarding agent baselines like ReAct and Reflexion,we agree that they can be helpful to determine feasible continuations. We have approximated this interaction by providing legal moves to the LLMs, simulating the effect of agents querying the environment for valid actions.  Unfortunately, even with this list of legal moves, the models performed poorly when the puzzle size was large.
>
> ### W3. Paper Refinement
> We appreciate your feedback on the presentation. We have carefully proofread the manuscript and made several refinements in the revised version.
>
> ### W4. Quantitative Analysis of Error Types
> In Section 4.5 (now Section 4.6 in the revision), we have conducted a detailed quantitative analysis of the error types encountered by all models. This includes metrics on the frequency of each error type and identification of the most common issues. The analysis reveals that reasoning errors are the most prevalent. Furthermore, even when models successfully complete puzzles, a significant proportion of their reasoning steps contain errors, highlighting a gap in their reasoning capabilities.
>
> ## Questions
> ### Q1. Evaluator System
> The term "Evaluator" refers to puzzle-specific evaluation functions that compute raw scores for each game. For instance, in the game Max Target, the evaluator calculates the coins collected by a player at each turn. These raw scores are subsequently normalized to a 0–1 range for cross-model comparison.
>
> ### Q2. Adversarial vs. Competitive Games
> The terms "adversarial games" and "competitive two-player games" are interchangeable in our benchmark. While two-player games can include cooperative scenarios, our benchmark focuses exclusively on competitive games, which we refer to as adversarial.
>
> ### Q3. Win Rate Computation in Figures 2 and 3
> In the tables on the right of Figures 2 and 3 (now Figures 2 and 29 in the revision), the win rate represents how frequently a model wins when competing against all other models. The corresponding figures display one-to-one win rates for each model pairing.
>
> ### Q4. Context Window Length
> We have included details about context window usage in Appendix C.3. Token consumption for gameplay varies, ranging from a few hundred tokens to approximately 64k tokens per game. However, multi-turn games significantly increase token usage cumulatively, with total consumption reaching several million tokens for games involving longer runs. While models can handle large context windows, the associated costs become prohibitively high in these scenarios.
>
> ### Q5. Table Reference Error
> Thank you for pointing out this error. We have corrected the reference in the revised manuscript (Section 4.1).

---

> ### Comment · Reviewer_X2oB · 2024-11-25
> **Further review**
>
> Thank you for the detailed response! I appreciate the efforts to address a lot of my concerns and have hence increased my score.
>
> > The term "Evaluator" refers to puzzle-specific evaluation functions that compute raw scores for each game. For instance, in the game Max Target, the evaluator calculates the coins collected by a player at each turn. These raw scores are subsequently normalized to a 0–1 range for cross-model comparison.
>
> Thanks for clarifying! It would be great if the authors could add this to the paper.
>
> > The terms "adversarial games" and "competitive two-player games" are interchangeable in our benchmark. While two-player games can include cooperative scenarios, our benchmark focuses exclusively on competitive games, which we refer to as adversarial.
>
> Thanks for clarifying. I think it would be better if the authors used "competitive 2-player" instead of "adversarial". The reason for this is that "adversarial" usually refers to something that's not only competitive/hard, but is so in an extreme way. Using this term here is confusing for people who might expect it to mean something more than "competitive 2-player".
>
> > In the tables on the right of Figures 2 and 3 (now Figures 2 and 29 in the revision), the win rate represents how frequently a model wins when competing against all other models.
>
> So is this a row-wise average of the matrix on the left?
>
> My main concern remaining with the paper is the evaluation section. I appreciate the authors' efforts to quantify the errors into different categories. However, it would be great if the authors could add clear criteria for when something is counted as a reasoning error vs. a memorization error or a comprehension error so that this becomes less subjective. These criteria could be added to the appendix if space is limited in the main paper. Finally, the writing still feels a bit rushed.

---

> > ### Author Response · Authors · 2024-11-28
> >
> > Thank you for your detailed response and constructive suggestions. I appreciate your reevaluation and thoughtful feedback.
> >
> > >Thanks for clarifying! It would be great if the authors could add this to the paper.
> >
> > Thank you for the suggestion. We already added this to the paper.
> >
> > >Thanks for clarifying. I think it would be better if the authors used "competitive 2-player" instead of "adversarial". The reason for this is that "adversarial" usually refers to something that's not only competitive/hard, but is so in an extreme way. Using this term here is confusing for people who might expect it to mean something more than "competitive 2-player".
> >
> > We appreciate your feedback and agree - we updated the terminology to "competitive 2-player" throughout.
> >
> > >So is this a row-wise average of the matrix on the left?
> >
> > Yes, you are correct - it represents the row-wise average.
> >
> > >My main concern remaining with the paper is the evaluation section. I appreciate the authors' efforts to quantify the errors into different categories. However, it would be great if the authors could add clear criteria for when something is counted as a reasoning error vs. a memorization error or a comprehension error so that this becomes less subjective. These criteria could be added to the appendix if space is limited in the main paper. Finally, the writing still feels a bit rushed.
> >
> >  Thank you for this helpful suggestion. We have given extensive definitions for each error type at Table 16 in the appendix.

---

> > > ### Comment · Reviewer_X2oB · 2024-11-30
> > > **Thank you.**
> > >
> > > I thank the authors for their receptiveness to feedback! I'm quite happy to see the changes that have been implemented, well done!
> > >
> > > While I'm tempted to increase my score further, the last thing holding me back is something mentioned as well by one of the other reviewers:
> > > > The writing of the paper seems to be rushed, and given the significant changes to the paper after the rebuttal, the paper may need a new round of peer-review. Therefore, I maintain my current score.
> > >
> > > I pretty much agree with this, and hence maintain my score for now as well.

---

### Official Review · Reviewer_GXDk · 2024-11-04

**Soundness:** 3
**Presentation:** 3
**Contribution:** 2
**Rating:** 5
**Confidence:** 3

**Summary:**

This paper provides a set of 24 puzzles benchmark for assessing the reasoning capabilities of LLMs. The puzzles cover a variety of task types such as single-player/multiplayer, adversarial, deterministic/stochastic, and two modalities of text-only and text-image. The authors then evaluate various open-source and proprietary LLMs, concluding that GPT4-o surpasses all other LLMs. The image-text puzzles also show the low performance of all LLMs on such tasks.

**Strengths:**

* The paper is well-written and easy to follow.
* Proposing benchmarks to asses the reasoning abilities of LLMs is an important task in the right direction.
* The puzzles support a high variety of possible situations.

**Weaknesses:**

* It is not clear how much utility can PuzzlePlex introduce in addition to the previous work SmartPlay (Wu et al., 2024). From what I understand, the main advantage of PuzzlePlex over SmarPlay is the presence of text-image puzzles and difficulty levels. But, the benchmark has only three image-text puzzles, all three of which are the visualized versions of text puzzles. In other words, the tasks are not inherently visual (e.g., like a task such as a jigsaw puzzle), but rather, a powerful enough OCR and a chain-of-thought prompt can bridge the gap between the visual puzzle and text puzzle.

* It is not clear how the puzzles are chosen. I understand the tasks are categorized into different natures. But, what capability is each puzzle measuring? (For instance, long context understanding? Spatial reasoning? Learning from interaction?). Such a categorization is present in SmartPlay while missing in PuzzlePlex.

**Questions:**

* Tables 4 and 5 have missing numbers. Is there any particular reason?
* In line 197, the reference to the Communications of the ACM is not correctly written.

---

> ### Author Response · Authors · 2024-11-24
>
> Thank you for your thoughtful feedback. I'm pleased to address each point below.
>
> ## Weaknesses
> ### W1. Utility of PuzzlePlex vs. SmartPlay
> When compared to SmartPlay, PuzzlePlex makes the following contributions:
>
> 1. **Broader Scope:** PuzzlePlex includes more games (24 versus 5 in SmartPlay).
>
> 2. **Difficulty Levels:** PuzzlePlex supports increasing  difficulty levels, making it possible to adapt the benchmark as LLMs improve.
>
> 3. **Adversarial Multi-turn Games:** Unlike SmartPlay, PuzzlePlex incorporates true adversarial multi-turn games that evaluate interactive planning and strategy. This is in contrast to Rock Paper Scissors in SmartPlay which is evaluated against a fixed-pattern program (e.g. cycling regularly between rock, paper, and scissor).
>
> Regarding the question of translating mage puzzles to text, we agree that OCR can extract state information.  However, we have found that text descriptions of visual puzzles often fail to induce LLMs effectively to reason about important constraints. For instance, in Sudoku, representing states with a visual structure may help LLMs identify constraints like ensuring unique values in each subgrid.
>
> ### W2. Puzzle Selection Criteria
> Our puzzle selection was guided by the goal of ensuring diversity across several dimensions:
>
> - **Game Types:** Single-player and two-player games, deterministic and stochastic games, single-turn and multi-turn scenarios.
> - **Rule Complexity:** Games with both strict and loose rules.
> - **Challenge Level:** We selected puzzles that are less common and therefore less likely to be present in LLMs' training corpora, making them more suitable for evaluating generalization and reasoning abilities.
>
> In Appendix A1, we show puzzles requiring comprehensive reasoning, while Appendix C3 illustrates diverse turn structures and context lengths, ranging from short to very long. The goal is to create a robust and evolving benchmark for LLM evaluation.
>
> ## Questions
> ### Q1. Missing Numbers in Tables 4 and 5
> The numbers are missing in Tables 4 and 5 (now Tables 14 and 15 in the revised version) because we evaluated only Pixtral and two Llama-3.2-Vision models on text-image puzzles, while GPT-4, Gemini-1.5-Pro, Gemini-1.5-Flash, and Claude-3.5-Sonnet were evaluated on both text and text-image puzzles.
> ### Q2. Incorrect Citation in Line 197
> Thank you for pointing this out. We have corrected these errors in the revised manuscript.

---

> > ### Comment · Reviewer_GXDk · 2024-11-25
> >
> > Thank you for your clarification and the new revision. I find it hard to detect the changes you have made during the rebuttal since the paper has grown from 34 pages before the rebuttal to 46 pages after the rebuttal. Several figures have been moved around, and some tables have been added that are not shown in blue.
> >
> > > Broader Scope: PuzzlePlex includes more games (24 versus 5 in SmartPlay).
> >
> > 24 games is not necessarily an advantage. (1) it could reduce the quality of games proposed in the benchmark (such as proper prompting for each game and ensuring its quality), (2) the experimental focus of your paper is mainly on a few tasks (such as Sudoku and SudoKill), and some results on aggregated games (such as deterministic, stochastic). For instance, the paper does not contain the results for all 24 individual tasks. (3) there are repetitive puzzles in the benchmark (e.g., 4 of the 24 tasks are sudoku variants).
> >
> > > Challenge Level: We selected puzzles that are less common and therefore less likely to be present in LLMs' training corpora, making them more suitable for evaluating generalization and reasoning abilities.
> >
> > I do not find this to be true; for instance, since the experimental focus of the paper is mainly on Sudoku tasks, any sufficiently capable LLM is able to write the C++ or Python code for solving a Sudoku puzzle.
> >
> > > The numbers are missing in Tables 4 and 5 (now Tables 14 and 15 in the revised version) because we evaluated only Pixtral and two Llama-3.2-Vision models on text-image puzzles, while GPT-4, Gemini-1.5-Pro, Gemini-1.5-Flash, and Claude-3.5-Sonnet were evaluated on both text and text-image puzzles.
> >
> > I suggest the authors have a more consistent evaluation system and avoid having missing numbers.
> >
> > Overall, although I find the adversarial tasks and playing of LLMs against each other to be interesting in this paper, the current additional utility of the benchmark compared to prior works seems to be very minimal. Also, the writing of the paper seems to be rushed, and given the significant changes to the paper after the rebuttal, the paper may need a new round of peer-review. Therefore, I maintain my current score.

---

> > > ### Author Response · Authors · 2024-11-28
> > >
> > > Thank you for your insightful response and suggestions.
> > >
> > > > I find it hard to detect the changes you have made during the rebuttal since the paper has grown from 34 pages before the rebuttal to 46 pages after the rebuttal. Several figures have been moved around, and some tables have been added that are not shown in blue.
> > >
> > > We apologize for this oversight and have now marked all new additions in blue for clarity.
> > >
> > > >24 games is not necessarily an advantage. (1) it could reduce the quality of games proposed in the benchmark (such as proper prompting for each game and ensuring its quality)
> > >
> > >  Thank you for this observation. While managing a large benchmark is challenging, we find value in the comprehensive testing it enables, helping identify specific areas for LLM improvement. However, we understand your concern about quality maintenance.
> > >
> > > >(2) the experimental focus of your paper is mainly on a few tasks (such as Sudoku and SudoKill), and some results on aggregated games (such as deterministic, stochastic). For instance, the paper does not contain the results for all 24 individual tasks. (3) there are repetitive puzzles in the benchmark (e.g., 4 of the 24 tasks are sudoku variants).
> > >
> > > You are correct that we have focussed on Sudoku/Sudokill as examples in the paper because Sudoku is familiar to many readers. The complete results will be included in future versions. You are also correct that some of the 24 puzzles are variants of one another. If we count the number of truly distinct puzzles, the count goes down to 11.
> > >
> > > >I do not find this to be true; for instance, since the experimental focus of the paper is mainly on Sudoku tasks, any sufficiently capable LLM is able to write the C++ or Python code for solving a Sudoku puzzle.
> > >
> > > Thank you for this insight. You're correct - while code generation for puzzles like Sudoku should be possible, current LLMs can only handle simpler versions.
> > >
> > > >I suggest the authors have a more consistent evaluation system and avoid having missing numbers.
> > >
> > > We agree that consistency is important. We plan to conduct comprehensive evaluations across all models for the final publication.
> > >
> > > >Overall, although I find the adversarial tasks and playing of LLMs against each other to be interesting in this paper, the current additional utility of the benchmark compared to prior works seems to be very minimal. Also, the writing of the paper seems to be rushed, and given the significant changes to the paper after the rebuttal, the paper may need a new round of peer-review. Therefore, I maintain my current score.
> > >
> > > We sincerely appreciate your detailed feedback and understand your position. Thank you for helping us improve the paper.

---

### Official Review · Reviewer_C9cj · 2024-11-06

**Soundness:** 2
**Presentation:** 2
**Contribution:** 2
**Rating:** 6
**Confidence:** 2

**Summary:**

The authors propose "PUZZLEPLEX", a benchmark that evaluates LLM reasoning and planning capabilities in a multi-turn adversarial environment. The authors also perform experimentation showing that LLM doesn't perform well in this benchmark.

**Strengths:**

1. The authors propose 24 parametrizable  puzzles
2. the authors propose experimentation on these puzzles, showing llm still can't solve them.

**Weaknesses:**

1. The binary/ternary scoring system (0,1 for single-player; 0,0.5,1 for adversarial) may be too simplistic and mask important performance nuances
2. No ablation studies to understand which aspects of the puzzles are most challenging for LLMs
3. No discussion of computational resources required or runtime comparisons
4. I would appreciate animation of the puzzle or visualization of it.

**Questions:**

1. Did you finetune the model on this benchmark? does fientuning help?
2. Did you provide a few-shot example when testing the evaluation? Does the performance improve when you provide some demonstration?
3. How would you expect LLM to solve it, do you think better planning technique would help?

---

> ### Author Response · Authors · 2024-11-24
>
> Thank you for your thoughtful feedback. I'm pleased to address each point below.
>
> ## Weaknesses
>
> ### W1. Simplistic Scoring System
> Thanks for your suggestion! In the revision, we introduce an additional metric, strength, derived from the Bradley-Terry model [1]. This metric allows us to unify comparisons across puzzle types and provides a more detailed perspective on model performance. The results are consistent with the original scoring system, showing that GPT-4o and Claude-3.5-Sonnet are the top-performing proprietary models, while Qwen2-72B is the best-performing open-source model.
>
> ### W2. Lack of Ablation Studies
> As in most ablation studies, we consider which factors contribute to the performance of a given LLM with respect to each puzzle. Two are most prominent: the size of the puzzle and the legality of moves. In section 4.4, we show that the size of the puzzle strongly influences the win ratio of the LLMs in two person games and in one person games – LLMs do better on smaller game sizes. We treat legality in section 4.5 where we show that simply giving the LLMs a set of legal moves increases their win ratio.
>
> ### W3. Lack of Discussion on Computational Resources
> Thanks for your suggestion. We have added a discussion on computational resources and runtime comparisons in Appendix B.3 of the revised paper.
>
> ### W4. Visualization of Puzzles
> Thank you for your suggestion. Upon publication we will provide a project page that includes visualization for each puzzle. This will enhance the understanding of the puzzles and make the benchmark more accessible to the community.
>
> ## Questions
> ### Q1. Fine-tuning on the Benchmark
> Thanks for your insightful question. We have not performed fine-tuning in this paper because we have focused on the creation of the benchmark. In future work, we plan to do tuning on reasoning steps as illustrated in Figure 30, where the red text reflects reasoning mistakes. Eventually, we believe LLMs will improve their performance on benchmarks like PuzzlePlex by fine tuning on reasoning steps.
>
> ### Q2. Few-shot Prompting
> We include results of one-shot prompting in Section 4.5. Our results show that it provides a performance boost only in scenarios where the state space is small. For larger puzzles, the benefits of one-shot prompting are minimal, suggesting (though not proving) that it is not a scalable solution for more complex tasks. This is a promising direction for future work.
>
> ### Q3. Better Planning Techniques
> We agree that better planning techniques would significantly help. Additionally, we believe two complementary approaches are important:
>
> 1. Agent-based methods: For example, a program could assist LLMs by checking the legality of moves, thereby eliminating the need for repeated reasoning about legality during every step.
>
> 2. Fine-tuning on reasoning steps: This approach would directly enhance the LLMs' ability to plan and reason effectively, as demonstrated by our error analysis in Section 4.6, which highlights the prevalence of flawed reasoning steps.
>
> [1] Roger R Davidson. On extending the bradley-terry model to accommodate ties in paired comparison experiments. Journal of the American Statistical Association.

---

### Official Review · Reviewer_STYd · 2024-11-09

**Soundness:** 2
**Presentation:** 2
**Contribution:** 2
**Rating:** 5
**Confidence:** 4

**Summary:**

The work introduces a new benchmark, puzzleplex, which helps evaluate LLMs' abilities in solving puzzle-based games. The benchmark covers 24 different puzzles for evaluation, and highlights its inclusion of multi-step adversarial reasoning games. The benchmark evaluate some popular LLMs in its experiments, and show that existing LLMs fail to play very well against rule-based baselines.
------------------------------------
Thanks authors for the response. I think the ToT complementary results are very illustrating; but it is vital to be implemented for all tested LLMs and datasets considering the community's wide recognition of the method for puzzle-solving games. However, the analysis from my view still lacks enough quantitative results, and the authors do not provide how fine-tuning would improve those open models. I believe the work could have more solid contributions by completing those details. So I prefer to maintain my score.

**Strengths:**

1. Good Motivation: Planning and reasoning are crucial for next-level LLMs. The benchmark could be timely for the community.
1. Systematic Benchmarking: I believe the benchmark provides useful resources for the community, and the results is of reference value to LLM development. I believe the authors have paid much efforts in building it.

**Weaknesses:**

1. Lack of analysis: While the construction of the benchmark is valuable, I think it is necessary for a benchmarking work to present sufficient analysis on its observation to guide later research and LLM training. However, the insights and analysis provided in this work is very limited, and the main observation of LLMs' lacking planning abilities in puzzle-solving is well-recognized in literature. I think authors should go deeper into the planning and reasoning behaviors of examined LLMs, providing quantitative insights into how LLMs could be improved,  strengths and shortcomings of representative specific LLM, and how open/proprietary LLMs differ.
2. Insufficient prompting strategy: As authors stated in related work, in Tree-of-Thought (ToT) the authors have identified that mere ReAct fails to reasonably represent LLMs' potential in solving puzzle-based games, and ToT could be necessary in such context. However, in PuzzlePlex, it is still the ReAct that is used for prompting rather than the ToT. As a result, I think the current conclusion in this work is of less referential value to community. I suggest authors to use ToT as a major prompting method here, which could significantly improve the referential value of insights in this paper.
3. Fail to show how to improve: As puzzle-based games have strong rule-based baselines, it is valuable to generate data in this way and train open LLMs on the problem to see if there is any reasonable improvement. The LLM community has witnessed too many prompting-based benchmarks in recent years but few endeavor to show how research and open-source community could catch up with proprietary ones.

Overall, I acknowledge the direction and problem the authors try to address, but I think the paper needs more contents and efforts to be qualified for ICLR.

**Questions:**

For adversarial multi-turn evaluation, there is a reference work AgentBench[1] in LLM Agent study, which includes an environment called Digital Card Game where evaluated LLMs need to play against rule-based methods and switch sides for each game. I think the authors should consider including the work as a reference.

[1] AgentBench: Evaluating LLMs as Agents (ICLR 24)

---

> ### Author Response · Authors · 2024-11-24
>
> Thank you for your thoughtful feedback. I'm pleased to address each point below.
>
> ## Weaknesses
>
> ### W1. Lack of Analysis
> Thank you for your suggestion. In the revised manuscript, We have added a quantitative error analysis in Sections 4.5 and 4.6, identifying common errors made by LLMs.
>
> We detail our findings as follows: **The most frequent issue is that they fail to follow the rules or fail to choose advantageous moves.** These issues become more prominent in puzzles with stricter rules. For example, in SudoKill, a player’s next move must be on a cell in the same row or column as the opponent's previous move, and the value chosen to fill in must comply with Sudoku rules. All the LLMs had trouble following that rule.  **Regarding different LLMs, we observed distinct patterns.** For instance, in board games, most LLM models fill in  cells sequentially, starting from the first empty cell in the top-left corner and progressing row by row, with Claude-3.5-Sonnet being an exception. For example, in Sudoku, Claude strategically fills cells that are easier to solve first, enabling subsequent moves to be more predictable and improving the overall problem-solving process. While proprietary LLM models (e.g. GPT-4o) generally outperform open-source models, **the open-source models demonstrate a better understanding of rules in some games.** For example, in the games of Max Target and Larger Target, only the Llama-3.1-405B model recognizes that the provided bags are randomized, which is illustrated in Figure 33.
>
> We think these insights can provide actionable guidance for future research and LLM training. Specifically, fine tuning models with scenarios that reward valid rule-following actions or penalize deviations may help models internalize stricter compliance. Additionally, incorporating curriculum learning approaches—starting with simpler puzzles and progressively introducing more complex ones—could improve both rule adherence and strategic decision-making over time. Our benchmark will thus encourage such improvements.
>
> ### W2. Insufficient Prompting Strategy
>
> **We have incorporated Tree-of-Thought (ToT) and one-shot prompting techniques in the revision (see Section 4.4).** Both approaches lead to performance improvements, with ToT showing more significant gains. However, the experimental results demonstrate that the effectiveness of ToT is closely related to the size of the state space: it provides significant benefits in smaller-scale scenarios (e.g. board sizes of 4x4 in the Sudoku game) but little benefit as the problem scale increases (e.g. when board sizes increase from 4x4 to 9x9). This shows that while existing advanced prompting methods can enhance performance, **they alone are insufficient to address the fundamental challenges posed by puzzle-solving tasks. We expect our benchmark to encourage research in this direction.**
>
> ### W3. Failure to Show How to Improve
> Our findings indicate that simply generating data is not an effective solution. Fine-tuning LLMs on state-change data alone does not enable them to generalize well to new puzzles. As discussed in the paper, the primary reasoning limitations of current LLMs remain: difficulties in identifying legal moves and making optimal decisions when faced with multiple candidate actions.
>
> To address these issues, we believe that adopting an agent-based approach represents an exciting direction for future research. For example, agents could be equipped with mechanisms (e.g., programmatic checks) to verify the legality of each move based on the game's rules. This verification can be reused across different turns, reducing the need for repeated reasoning about legality for every candidate move. Additionally, fine-tuning directly on reasoning steps—not just state changes—will enable LLMs to improve their planning and reasoning capabilities.
>
> Moreover, our error analysis in Section 4.6 reveals that a significant proportion of reasoning errors occur even when the final results are correct by chance, because the moves decided by the LLMs cannot be supported by their reasoning steps. Improving the quality and reliability of reasoning steps is critical for better puzzle-solving and represents a promising direction for our future work.
>
> ## Question
>
> Thank you for the suggestion. We have included the AgentBench paper as a reference in our revised manuscript.

---

### Meta-Review · Area_Chair_GQcb · 2024-12-18

**Metareview:**

The paper introduces a benchmark designed to evaluate the reasoning and planning capabilities of large language models (LLMs) in puzzle-solving tasks. Experiments with popular LLMs indicate significant limitations in reasoning and planning, with GPT-4 outperforming others in some cases but still falling short compared to rule-based baselines.

The reviewer agreed that planning and reasoning capability in LLMs is a timely topic, and that providing systematic and diverse set of evaluation tasks is important. However, the utility of PuzzlePlex compared to existing benchmarks like SmartPlay is not significant (other than the adversarial multi-turn design), and many puzzles are slight variants of Sudoku. The reviewers noted that the paper seems rushed with outdated prompting strategies and too much has changed during the rebuttal phase. The proposed benchmark also offers very limited insights other than LLMs falling short in succeeding at these games. I agree with all concerns raised by the reviewers and recommend the authors to iterate on the work to improve the practicality and insight of the benchmark.

**Additional Comments On Reviewer Discussion:**

The reviewers raised many concerns such as value of the new benchmark given existing similar benchmarks. The overall sentiment post rebuttal is that the writing of the paper seems too rushed and the paper needs a new round of peer-review.

---

### Decision · Program_Chairs · 2025-01-22

Reject